# Forest impacts on snow accumulation and ablation across an elevation gradient in a temperate montane environment

Travis R. Roth[1], Anne. W. Nolin[1]

[1]Water Resource Sciences, Oregon State University, Corvallis, OR, 97331, USA

5    *Correspondence to*: T. R. Roth (rothtra@science.oregonstate.edu)

**Abstract.** Forest cover modifies snow accumulation and ablation rates via canopy interception and changes in sub-canopy energy balance processes. However, the ways in which snowpacks are affected by forest canopy processes vary depending on climatic, topographic and forest characteristics. Here we present results from a 4 year study of snow-forest interactions in the Oregon Cascades. We continuously monitored snow and meteorological variables at paired forested and open sites at three elevations representing the Low, Mid, and High seasonal snow zones in the study region. On a monthly to bi-weekly basis, we surveyed snow depth and snow water equivalent across 900 m transects connecting the forested and open pairs of sites. Our results show that relative to nearby open areas, the dense, relatively warm forests at Low and Mid sites impede snow accumulation via canopy snow interception and increase sub-canopy snowpack energy inputs via longwave radiation . Compared with the Forest sites, snowpacks are deeper and last longer in the Open site at the Low and Mid sites (4 – 26 days and 11 – 33 days, respectively). However, we see the opposite relationship at the relatively colder High sites with the Forest site maintaining snow longer into the spring by 15 – 29 days relative to the nearby Open site. Canopy interception efficiency ($C_{IE}$) values at the Low- and Mid-Forest sites averaged 79 % and 76 % of the total event snowfall, whereas $C_{IE}$ was 31 % at the lower density High-Forest site. At all elevations, longwave radiation in forested environments appears to be the primary energy component due to the maritime climate and forest presence, accounting for 82 %, 88 %, and 59 % of total energy inputs to the snowpack at the Low-, Mid-, and High-Forest sites, respectively. Higher wind speeds in the High-Open site significantly increase turbulent energy exchanges and snow sublimation. Lower wind speeds in the High-Forest site create preferential snowfall deposition. These results show the importance of understanding the effects of forest cover on sub-canopy snowpack evolution and highlight the need for improved forest cover model representation to accurately predict water resources in maritime forests.

25    **1 Introduction**

Snowpacks the world over are changing. Increasing global temperatures and accompanied climatic changes are altering snowpack characteristics and shifting melt timing earlier (McCabe and Clark, 2005; Mote, 2006; Mussleman et al., 2017). The timing, intensity, and duration of snowmelt depends on climatic and physiographic variables. In the topographically diverse western U.S. the distribution of snow cover is governed by regional climate, elevation, vegetation

presence/absence, and forest structure (Elder et al., 1998; Harpold et al., 2013). Forests overlap with mountains across this region and modify snow accumulation and ablation rates through canopy interception and a recasting of the sub-canopy energy balance (Hedstrom and Pomeroy, 1998; López-Moreno and Stähli, 2008; Varhola et al., 2010). Recently, a considerable amount of effort has been expended in research into the snow-forest processes that control the distribution of

snow in mountainous regions (Stähli and Gustafsson, 2006; Jost et al., 2007; López-Moreno and Latron, 2008; Musselman et al., 2008; Ellis et al., 2013; Moeser et al., 2015). While these studies have focused on cold, predominately continental snowpacks few have investigated snow-forest process interaction in warm maritime environments where snow is especially sensitive to changes in energy balance (Storck et al., 2002; Lundquist et al., 2013). Maritime snowpacks accumulate and reside at temperatures near the melting point. Such snowpacks do not fit the simple accumulation-ablation model of a

monotonic increase until peak snow water equivalent (SWE) followed by a monotonic decrease to snow disappearance. Such temperature sensitive snowpacks may experience disproportionate effects of climate warming and changing forest cover (Nolin and Daly, 2006; Dickerson-Lange et al., 2015). Ramifications of these impacts have far reaching eco-hydrological impacts across the snowmelt dependent western U.S., highlighting the continued need for research into snow-forest process interactions in maritime montane settings (Mote, 2006; Harpold et al., 2015; Vose et al., 2016).

In the Pacific Northwest, United States (PNW), mountain environments are a disparate composite of forest cover driven by forest harvest, regrowth, and natural disturbance. Forest disturbance can have significant impacts on snow processes whose effects can range from immediate (Boon, 2009) to decadal (Lyon et al., 2008; Gleason and Nolin, 2016). At the stand scale, forests attenuate wind speeds thereby suppressing turbulent mixing of the near surface atmosphere (Liston and Sturm, 1998); modify the radiation received at the snow surface through shifts in shortwave and longwave contributions

and reduced surface albedo (Sicart et al., 2004; O'Halloran et al., 2012; Gleason et al., 2013); and temporally shift seasonal and event scale accumulation and ablation patterns through canopy snowfall interception (Varhola et al., 2010). Natural and anthropogenic alterations in forest cover such as mountain pine beetle infestation, forest management practices, and forest fire affect snow processes by modifying forest structure, i.e. canopy cover and gap size (Boon, 2009; Bewley et al., 2010; Ellis et al., 2013) and snow albedo (Gleason et al., 2013; Gleason and Nolin, 2016). The frequency and intensity of forest

fires has been increasing (Westerling et al., 2006; Miller et al., 2009; Spracklen et al., 2009) impacting accumulation and ablation rates (Gleason et al., 2013) and is anticipated to continue increasing (Moritz et al., 2012; Westerling et al., 2011). While prolonged droughts, and a future of increasing drought prevalence, have increased water stress creating changes in forest characteristics across the western U.S. (Allen, 2010; Choat, 2012; Dai, 2013). Disturbances of this type alter the snow-forest dynamic through a modification of the magnitudes of central process relationships, often resulting in unanticipated

outcomes (Lundquist et al., 2013). The present reality and spectre of continued future change to climate and forest cover underscores the increasing importance of characterizing vegetation impacts on snow accumulation and ablation within warm, topographically varied terrains.

Elevation (as a proxy for temperature) and forest canopy cover are important controls on peak snow accumulation (Geddes et al. 2005; Jost et al., 2007). Elevation drives snow accumulation and is the principle predictor of peak snow water

equivalent (Gray 1979; Elder et al., 1991; Sproles et al., 2013). The partitioning of precipitation between rainfall and snowfall is determined by atmospheric temperature and the elevation of the rain-snow transition can be described as a function of the temperature lapse rate. Forest canopies intercept snow, reducing sub-canopy accumulation (Schmidt and Gluns, 1991; Hedstom and Pomeroy, 1998; Musselman et al., 2008). The magnitude and rate of canopy interception is also affected by air temperature. Air temperature has been shown to have an inverse relationship with canopy interception (Andreadis et al., 2009) and a non-linear correlation with event size (Hedstrom and Pomeroy, 1998), these relationships are often based on few measurements and at a single point. Forests also reduce solar radiation reaching the snowpack surface (Link and Marks, 1999; Hardy et al., 2004) and increase longwave radiation at the snowpack surface (Lundquist et al., 2013) thus modifying net radiation (Sicart et al., 2004). Forest cover reduces wind speed thereby reducing latent and sensible heat flux at the snowpack surface (Link and Marks, 1999; Boon, 2009). The direct effect of wind speed on canopy snow interception has not been explicitly studied, with most research focusing on wind redistribution of snow (Gary, 1974; Pomeroy et al., 1997; Liston and Sturm 1998; Woods et al., 2006). Research demonstrates that forests reduce wind speed and can lead to increased snow accumulation in canopy gaps or forest clearcuts where wind speeds decline and snow is released from upwind canopy flow (Gary, 1974). These combined forest effects on sub-canopy energy and mass balance can accelerate or delay the onset and rate of snowmelt (Varhola et al., 2010). These studies highlight the key differences between forested and open areas, and the effects of elevation on snowpack evolution. With strong agreement that the western U.S. will be facing warmer winters in the future and new understanding that snow in forested regions is more sensitive to increased temperatures than snow in non-forested regions (Lundquist et al., 2013) it is critical that we measure, characterize, and understand maritime snow-forest interactions. This study examines and evaluates the combined effects of forest cover, climate variability, and elevation on snow accumulation and ablation in a maritime montane environment. Specifically, we focus on the following research questions:

1)      To what extent do forests modify snow accumulation and ablation in a maritime temperate forest?

2)      How does canopy interception affect sub-canopy snowpack evolution across an elevation gradient?

3)      How does forest cover affect the sub-canopy snow surface energy balance relative to adjacent open areas and what are the principal drivers of melt?

In subsequent sections, we describe the study area; present research methods for field measurements, energy balance calculations, and snow modelling; present our key findings; and, conclude with a description of potential applications and future steps.

## 2 Methods

### 2.1 Description of the Study Area

The McKenzie River Basin (MRB) is part of the greater Willamette River Basin in western Oregon, USA (Fig. 1). It covers an area of 3041 km² and spans an elevation range from 150 m to over 3100 m at the crest of the Cascades

Mountains that flank its eastern boundary. Orographic uplift results in average annual precipitation ranging from 1000 mm at lower elevations to over 3500 mm at the highest elevations in the basin (Jefferson et al., 2008). The rain-snow transition zone sits between 500 – 1200 m (Marks et al., 1998). The area above the transition zone accounts for 12 % of the total area with the Willamette River Basin, yet contributes 60 – 80 % of summer baseflow to the Willamette River (Brooks et al., 2012). The MRB elevation between 1000 and 2000 m is especially important as it comprises 42% of the total area within the MRB and snowmelt from this elevation band accounts for nearly 93 % of the total snow water storage (Sproles et al., 2013). Warm snowpack conditions facilitate frequent melt events during the winter months of December, January and February (DJF), commonly masking the distinction between accumulation and ablation periods. Nolin and Daly (2006) showed that snowpack in this region has an acute sensitivity to temperature with the low elevation snow zones of the Oregon Cascades classified as the most 'at-risk' snow within the region. The Natural Resources Conservation Service (NRCS) has been monitoring seasonal snowpack within the MRB since the early 1980's by a point-based snow telemetry (SNOTEL) network. Placement of SNOTEL stations was designed to be representative of water producing regions of a watershed and yet network stations were ultimately placed in protected, accessible locations (Molotch and Bales, 2006). However, the limited configuration was not designed to understand forest-snow processes nor with future climate change in mind and therefore a statistically unbiased approach to site selection that is spatially representative is needed for any substantial snow observation network (Molotch and Bales, 2006). This underscores the need for intelligent and statistically relevant snow monitoring sites that go beyond the existing network. Section 2.2 outlines snow monitoring network we deployed in water year (WY) 2012 that meets these stated needs.

### 2.2 The Oregon ForEST network

The Oregon Forest Elevation Snow Transect (ForEST) network extends from the rain-snow transition zone through the seasonal snow zone in the Oregon Cascades with paired forested and open sites at three elevations, Low (1150 m), Mid (1325 m) and High (1465 m) (Fig. 1). The ForEST network was designed to efficiently represent the range of peak SWE within the basin. Using a binary regression tree (BRT) approach, we identified elevation, vegetation type and vegetation density as the key predictor variables and we used them to classify the basin and locate our network sites (Molotch and Bales, 2006; Gleason et al., 2017). At each of three elevation zones, we established Open (low forest density) and Forest (high forest density) site pairs in adjacent areas, while controlling for slope and aspect. Open sites consisted of < 20 % canopy cover while corresponding Forest sites had > 60 % canopy cover based on the 2001 National Vegetation Cover Database (Homer et al., 2007) and subsequently verified by *in situ* measurements.

At each of the six sites within the ForEST network tower-based instruments continuously measured snow depth, incoming and reflected shortwave radiation, air temperature, relative humidity, wind speed, wind direction, and soil temperature and soil moisture (Table S1). Sensor measurement frequency was 15 sec with output values averaged over a 10 min period. The suite of sensors allowed the calculation of the snow surface energy balance through either direct measurement, e.g. solar radiation, or through empirical equations, e.g. turbulent fluxes, longwave radiation. The sSnow-

climate monitoring stations were deployed and active for the duration of the snow season at all sites, typically from mid-late November thru May, with minimal disruptions due to battery or mechanical failures. We present results from the Low and Mid sites for WY2012 – WY2015 and results from the High sites which were added to the network for WY2014 and WY2015.

5        Additionally, SWE and snow depth measurements were collected along 900 m transects ("snow courses") extending from the forested to the open sites in the Low, Mid, and High elevation zones. SWE measurements locations were restricted to > 50 m from the forest edge to eliminate canopy edge effects. These snow course surveys were conducted on a monthly basis during the accumulation period then bi-weekly during the ablation phase until the snow disappearance date (SDD). SWE was measured using a snow tube ("Federal sampler") and snow depth was measured using a steel probe pole. Within 10      each vegetation cover type, e.g. Open or Forest site, SWE measurements were made at 100 m intervals with snow depth measurements every 5 m. Snow course data used in this analysis are from WY 2012 – WY 2015 for all ForEST network sites. To estimate SDD for each site we calculated the snowpack ablation rate using median snow depths from the last two snow courses of the season and linearly extrapolating to the date of zero snow depth. SDD represents the date when the primary seasonal snowpack disappears and doesn't take into account late season periods of accumulation/ablation. We 15      excluded data from the historically low WY 2015 due to a near absence of winter snow.

### 2.3 Canopy Interception Efficiency

        Forest structure characteristics at each site were quantified using ground-based conventional forest inventory methods. At transect locations coinciding with SWE measurements, individual tree characteristics were measured within each quadrat and averaged for that particular site, i.e. diameter at breast height (DBH), crown radius, tree height, and tree 20      species (Table 1). Forest density was performed using a plotless density estimator approach described in Elzinga et al., (1998). Forest canopy at each site was further characterized using skyward looking hemispherical photographs acquired using a Nikon Coolpix 990 digital camera equipped with a FC-E8 fisheye converter, which has a 180° field-of-view (Inoue et al., 2004). The hemispherical photographs were assessed with Gap Light Analyzer 2.0 to measure leaf area index (LAI) and canopy closure (CC), which is the complement of the sky view fraction (Frazer et al., 1999).

25      During the snow accumulation period forest canopy plays a large role in reducing snowpack by intercepting incoming snowfall, prohibiting a significant portion from accumulating on the forest floor. A forest canopy is the integrated sum of the forest overlaying the ground surface, this includes needles, leaves, branches, and trunks. The canopy structure is the primary control on canopy interception followed by event specific variables i.e. event size, air temperature, and wind speed (Varhola et al., 2010). Canopy snow interception is inherently difficult to accurate quantify due to the temporally 30      sensitive impacts of local climate on the canopy itself and the limited measurement capabilities to directly measure canopy interception (Martin et al., 2013; Friesen et al., 2014). From measured snowfall at each climate station within the ForEST network we calculated percent canopy interception efficiency ($C_{IE}$) for daily snowfall events. A snowfall event is defined as the daily increase in measured snow depth in the Open sites greater than 3 cm. Ryan et al., (2008) showed that acoustic snow

depth measurement error for the Campbell Scientific SR50a is $\pm$ 2 cm under normal field conditions. Therefore, to reduce the influence of depth measurement error on our snow event classification we used a $\geq$ 3 cm threshold for our analysis. $C_{IE}$ is calculated as:

$$C_{IE} = \left[\frac{O_S - F_S}{O_S}\right] \times 100. \tag{1}$$

5    where, $O_S$ and $F_S$ are the measured snowfall (cm) in the Open and Forest sites, respectively. $C_{IE}$ was calculated for individual events and for seasonal averages at each Forest site.

## 2.4 Snow Surface Energy Balance

A snow surface energy balance was calculated at a daily time step using aggregated 10 min meteorological measurements from each site. Each energy balance component was either directly measured or calculated using empirically 10    derived equations valid for a maritime snowpack. Total energy into the snowpack equals the combined incoming and outgoing energies experienced at the surface of the snowpack. The governing equation for the snow surface energy balance is:

$$\Delta Q = Q_{SR} + Q_{LW} + Q_E + Q_H + Q_C \tag{2}$$

where, $\Delta Q$ is the change in total energy present at the snow surface (W m$^{-2}$); $Q_{SR}$ is total solar radiation (W m$^{-2}$); $Q_{LW}$ is total 15    longwave radiation (W m$^{-2}$); $Q_E$ is latent heat (W m$^{-2}$); $Q_H$ is sensible heat (W m$^{-2}$); $Q_C$ is conductive energy (W m$^{-2}$).

A critical component within the snow surface energy balance calculations is the determination of the snow surface temperature, $T_{snow}$ (Andreas, 1986). $T_{snow}$ controls directional energy flows by regulating temperature and vapor flux gradients between the atmosphere and the snowpack, which control the sensible and latent heat transfer, respectively. $T_{snow}$ is also the primary control of longwave radiation emitted from the snowpack. However, $T_{snow}$ is difficult to directly measure 20    and is therefore estimated as function of the dewpoint (frostpoint) temperature, $T_{dew}$, as demonstrated by Raleigh et al., (2013). Using $T_{dew}$ to estimate daily averages of $T_{snow}$ reduces bias and is a reasonable first order approximation at standard height measurements (Raleigh et al., 2013).

## 2.4.1 Solar Radiation

Incoming and reflected solar radiation were each measured using an upward facing and downward facing LI-200s™ 25    pyranometer (LI-COR). The pyranometers have a spectral range of 400 – 1100 nm and a field of view of 180°. Net solar radiation is calculated as:

$$Q_{SR} = SR_{in} \times (1 - \alpha) \tag{3}$$

where, $SR_{in}$ equals the measured incoming shortwave radiation (W m$^{-2}$). Albedo, $\alpha$, was calculated as the ratio of reflected and incoming measured solar radiation. When periods of newly fallen snow obscured the upward facing solar pyranometer,

i.e. when $\frac{SR_{out}}{SR_{in}} > 1$, a value of $\alpha = 0.9$ was used. Similarly, when $\frac{SR_{out}}{SR_{in}} < 0.3$, a value of $\alpha = 0.3$ was used to adequately simulate the lower bound of forest floor albedo during the ablation period (Melloh et al., 2002).

### 2.4.2 Longwave Radiation

Longwave radiation is rarely directly measured in the seasonal snow zone both due to high cost in both absolute,
e.g. instrument cost, and relative terms, e.g. energy requirements. Longwave radiation balance was calculated as:

$$Q_{LW} = L{\downarrow} + L{\uparrow} \tag{4}$$

where, $L{\downarrow}$ is the calculated longwave radiation received by the snowpack surface and $L{\uparrow}$ is the calculated longwave radiation emitted by the snow surface. Longwave radiation emitted at the snow surface is approximated by:

$$L{\uparrow} = \varepsilon_{snow}\,\sigma\,T_{snow}^4 \tag{5}$$

where, $\varepsilon_{snow}$ is the snow surface emissivity and is set at 0.96 (Link and Marks, 1999).

A variety of empirically derived formulas exist for calculating incoming longwave radiation under clear ($LW_{clear}$) and cloudy skies at various sites throughout the world (Brutsaert, 1975; Sicart et al., 2004; Flerchinger et al., 2009). All derivations are variations of the general form of the Stefan-Boltzmann equation that relates clear sky incoming longwave radiation to atmospheric emissivity ($\varepsilon_{clear}$), the Stefan-Boltzman constant ($\sigma$), and air temperature $T_{air}$ (K).

$$LW_{clear} = \varepsilon_{clear}\,\sigma\,T_{air}^4 \tag{6}$$

Many of these parameterizations are site specific or do not incorporate a cloud cover component nor account for longwave radiation emitted from the canopy (Hatfield et al., 1983; Alados-Alboledas et al., 1995). The presence and type of cloud cover affects how longwave radiation is absorbed and transmitted through the atmospheric air column, significantly affecting emissivity, and subsequently the magnitude of incoming longwave radiation (Sicart et al., 2004, Lundquist et al., 2013).
Incorporating a sky-view factor (SVF) into the longwave radiation calculations allowed us to partition the incoming longwave into atmospheric and forest canopy contributions.

Following Flerchinger et al., (2009) we performed a comparative analysis of various longwave radiation algorithms and measured net longwave radiation. Table S2 shows two clear sky algorithms and three cloud correction algorithms used in the comparison, totalling six combinations in all, with the "best-fit" algorithm determined by root means squared error
(RMSE). We measured longwave radiation using a Huskeflux NR1 net radiometer during spring 2013 for a 2 week period in a forested site within the MRB (Gleason et al., 2013) and for a 10 day period in an adjacent open area., excluding a 4 day period of rain. The NR1 measures four separate components of the surface radiation balance, separately measuring incoming and reflected solar radiation and both incoming and outgoing far infra-red radiation. The pyrogeometers have a built in Pt100 temperature sensor for calculation of both the sky and surface temperature. Additionally, they are heated, with temperature
compensation, to avoid moisture build up on the thermopile sensors. The predicted incoming longwave radiation results of each method were then compared to the NR1 measured incoming longwave radiation using RMSE, Table S3. We found that the best approximation for incoming longwave energy was the clear sky algorithm of Dilley and O'Brien (1998) combined

with the cloud adjustment of Crawford and Duchon (1999). The combined Crawford-Dilley method was therefore used in all longwave calculations going forward and is calculated as:

$$L\downarrow = (SVF) * \varepsilon_{adj} \, \sigma \, T_{air}^4 + (1 - SVF) * \varepsilon_{snow} \, \sigma \, (T_C^4) \tag{7}$$

where, $SVF$ is the sky view factor and represents the fraction of viewable sky from the perspective of the ground surface; $\varepsilon_{adj}$ is the adjusted atmospheric emissivity; and $T_c$ is the temperature of the forest canopy (K). $T_c$, is highly variable and typically not directly measured. Literature suggests a range of temperature of an increase of $4 - 30$ K from measured air temperature (Derby and Gates, 1966; Pomeroy et al., 2003; Essery et al., 2008). We assumed canopy temperature to be equal to $T_{air} + 4$ K based on Boon (2009). Adjusted emissivity accounts for changes in atmospheric emissivity due to cloud cover and is found by adjusting the clear-sky emissivity ($\varepsilon_{clear}$) by some estimation of cloud cover. The Dilley and O'Brien (1998) clear sky algorithm is as:

$$L_{clear} = 59.38 + 113.7 \times \left(\frac{T_{air}}{273.16}\right)^6 + 96.96 \times \sqrt{\frac{w}{25}}$$

(8)

$$w = \frac{465 \times \frac{e_o}{100}}{T_{air}} \tag{9}$$

The Crawford and Duchon (1999) cloud correction adjusted algorithm requires $\varepsilon_{clear,}$ which we computed from Eq. (8) and is in the following form:

$$\varepsilon_{adj} = (1 - s) + s \times \varepsilon_{clear} \tag{10}$$

where, $s$ is the solar ratio, an approximation of cloud cover, and is equal to the ratio of measured incoming solar radiation and potential solar radiation (Lhomme et al., 2007).

### 2.4.3 Turbulent Heat Flux

The turbulent fluxes of latent and sensible heat are calculated using indirect methods. Latent heat exchange was calculated using the method found by Kustas et al. (1994)

$$Q_E = \left(\rho_a \, 0.622 \, \frac{L}{P_a}\right) C_e \, U_Z \, (e_a - e_0) \tag{11}$$

where $\rho_a$ is the density of air (kg m$^{-3}$), $L$ is the latent heat of vaporization or sublimation (J kg$^{-1}$), $P_a$ is the total atmospheric pressure (Pa), $C_e$ is the bulk transfer coefficient for vapor exchange, $U(z)$ is the wind speed at height $Z$ (m) above the snow surface (m s$^{-1}$), $e_a$ is the atmospheric vapor pressure at height $Z$ above snow surface (Pa), and $e_o$ is the vapor pressure at the snow surface (Pa). This calculation favors the bulk aerodynamic approach adapted from Brutsaert (1982), as direct measurement is limited and successful implementation difficult in remote environments (Moore, 1983; Marks and Dozier, 1992; Marks et al., 1998). $C_{en}$ is the bulk transfer coefficient for vapor exchange under neutral stability and calculated as:

$$C_{en} = k^2 \left[ln\left(\frac{z}{z_0}\right)\right]^{-2} \tag{12}$$

where, $k$ is von Karman's constant 0.4 (-) and $Z$ is the height of the measurement above the snow surface (m) and was 3 m above the snow free ground surface for the Low and Mid sites and 4.5 m for the High sites. Additionally, the surface roughness length, $Z_0$ is a primary control on the bulk transfer coefficient, Eq. (12). The roughness length is affected by snow properties and is generally found to have values ranging from 0.001 – 0.005 m (Moore, 1983; Morris, 1989). This value

represents the mean height of snow surface obstacles that impede air movement over the snow surface. In our analysis we used a median value, 0.003 m, due to the variable nature of the seasonal snowpack.

The bulk aerodynamic approach is guided by stability conditions in the air above the snow surface. The stability of the air column is determined by application of the dimensionless bulk Richardson number ($Ri_B$) which relates the density gradient to the velocity gradient, in this case the energy of buoyancy forces to the energy created by shear stress forces. $Ri_B$

is calculated as:

$$Ri_B = \frac{g\ Z\ (T_{air} - T_{snow})}{0.5\ (T_{air} + T_{snow})\ U\ (z)^2} \tag{13}$$

where, $g$ is the acceleration due to gravity, 9.81 m s$^{-2}$. As Eq. (13) shows, the stability of the atmosphere is temperature dependent. Under stable conditions where the relatively warm air column settles over the snow surface will cool and become dense, impeding turbulent mixing. Conversely, when the air column is relatively colder than the snow surface free

convection of the air column exists where the air warms and expands causing increased mixing and unstable conditions. Positive values of $Ri_B$ indicate stable conditions whereas negative values indicate instability. Corrections for atmospheric stability effects are inconsistent within the literature and therefore remain an area of continued study (Anderson, 1976; Oke, 1987; Kustas et al., 1994; Andreas, 2002). In this study we employ Eq. (14a and 14b) as the general stability correction equations (Oke, 1987)

$$Unstable: \frac{C_e}{C_{en}} = (1 - 16Ri_B)^{0.75} \tag{14a}$$

$$Stable: \frac{C_e}{C_{en}} = (1 - 5Ri_B)^2 \tag{14b}$$

Sensible heat exchange, much like latent heat exchange is controlled by temperature, wind speed, roughness length, and atmospheric stability conditions. Sensible heat flux was calculated as:

$$Q_H = \rho_a\ C_p\ C_h\ u_a\ (T_{air} - T_{snow}) \tag{15}$$

where, $C_p$ is the specific heat of dry air (J kg$^{-1}$ K$^{-1}$) and $C_h$ is the bulk transfer coefficient for sensible heat. Here we assumed $C_e = C_h$ and $C_{en} = C_{hn}$.

## 3 Results

### 3.1 Snow Surveys

Values for April 1 SWE, as calculated from the NRCS SNOTEL stations, range from 9 % (WY 2015) to 139 %

(WY 2012) of the 30-year median reference period (1981 – 2010). Snow surveys conducted at the Low and Mid elevation

sites for WY 2012 – 14 show SWE at the Open site to be consistently greater and snow cover lasting longer into the spring than the adjacent Forest site (Fig. 2). During the average snow year of WY 2013 (93 % of 30 year median) the Low and Mid sites showed substantial differences between Open and Forest SWE throughout the accumulation and ablation season, whereas at the High sites SWE amounts were similar in the Open and Forest. Conversely, snow lasted longer into the spring in the High-Forest site relative to the High-Open site. Because April 1 SWE may not accurately represent annual peak SWE at low and mid elevations within the PNW, we use the date of peak SWE in the following analysis. Therefore, peak SWE at the Low-Open site was 209 %, 215 %, 225 %, and 242 % of the Forest site peak SWE, respectively for WY 2012 – WY 2015. Peak SWE at the Mid-Open site was 200 %, 280 %, 328 %, and 302 % of the Forest site peak SWE, respectively for WY 2012 – WY 2015. However, SWE at the High-Forest site is consistently higher than at the High-Open site, 111 %, 103 %, 125 %, and 110 % for WY 2012 – WY 2015, respectively.

Excluding the historically low snowpack of WY 2015 (Sproles et al., 2017), the three-year average snow depth ablation rates in the Forest sites at Low and Mid elevation were 1.3 and 1.2 cm d$^{-1}$ while the Open sites was 4.1 and 3.1 cm d$^{-1}$, respectively (Table 2). Melt rates at the High site were greater at both sites than their lower elevation counterparts, with a rate of 4.7 cm d$^{-1}$ at the High-Open site and a rate of 3.2 cm d$^{-1}$ for the High-Forest site. At the Mid-Open snow persistence exceeds that of the Mid-Forest site by 11 – 33 days. This is a similar finding to the low elevation sites where snow lasted longer at the Low-Open by 4 – 26 days compared with the Low-Forest site. Conversely, the High-Forest site maintains snow longer into the spring by 15 – 29 days when compared to the High-Open site.

**3.2 Forest characteristics and canopy interception efficiency**

Results show that $C_{IE}$ in the Low- and Mid-Forest sites, for WY 2012 – WY 2015, were 79 % and 76 % of the total event snowfall, whereas $C_{IE}$ was 31 % at the High-Forest site (Table 2). $C_{IE}$ showed no significant threshold behavior between event size and $C_{IE}$, although there is an inverse relationship between duration and $C_{IE}$ at the Low and Mid sites. Events that lasted for a single day had an average canopy interception efficiency of 87 % with a reduction in average $C_{IE}$ with increasing event length, from 73 % for a two-day event, 57 % for a three-day event, to 51 % for any event lasting longer than four days. Due to the low snow years of WY 2014 and WY 2015 the High site had only four events that lasted longer than one day and therefore no relationship with event duration could be identified. Using event based $C_{IE}$ for all snowfall events we calculated how much snow was removed by the canopy at each elevation and compared that with each event snowfall amount (Fig. 3). The Low elevation site has a high correlation between $C_{IE}$ and event size for all qualifying events ($R^2 = 0.86$) and an estimated overall snow removal efficiency of 58 %. The Mid elevation site has a lower correlation ($R^2 = 0.64$) between $C_{IE}$ and event size and an overall snow removal efficiency of 42 %. The linear relationship of the Low and Mid sites is similar to what Storck et al. (2002) found for single Douglas-fir (*pseudotsuga menziesii*) over a 2 year study in Oregon, that 60 % of event snowfall was intercepted by the canopy. This relationship does not hold at the High elevation site with an overall snow removal efficiency of only 4 %. We note an apparent threshold behavior where events less than 15 cm have a stronger linear relationship between event size and $C_{IE}$ (Fig. 3) and the canopy was more effective at snow removal

for events in that range compared with events greater than 15 cm. For events < 15 cm, canopy removal rates increase to 88 % for the Low site, 89 % for the Mid site, and interestingly, a weak correlation emerges, $R^2$ of 0.27, with 50 % removal for the High site.

### 3.3 Energy balance

To better understand the energy balance effect of forest canopies on snow accumulation and ablation we calculated the mean daily energy balance components for the Low and Mid elevation sites for WY 2012 – WY 2015 and for WY 2014 and WY 2015 for both High elevation sites (Fig. 4). Net radiation is the major component at all sites while the turbulent fluxes, sensible and latent heat, are only significant at the High-Open site. Turbulent fluxes at all other sites are only episodically important and do not account for any significant amount of energy at the monthly or annual timescales. On an

annual basis, shortwave radiation is the primary component of the energy balance at all Open sites whereas longwave radiation dominates at all Forest sites. There is a strong dominance of shortwave (longwave) energy at the Low and Mid-Open (Forest) site, where it accounts for 89 % and 71 % (93 % and 92 %) of the average annual net energy balance, respectively. While few studies in maritime forested environments on the energy balance exist there is evidence of longwave radiation as the dominating term during rain on snow (ROS) events within forests (Berris and Harr, 1987; Mazurkiewicz et

al., 2008; Garvelmann et al., 2014). Berris and Harr (1987) showed that longwave radiation accounted for 38-88% of all ROS event snowmelt. Garvelmann et al., (2014) found that in two ROS events longwave accounted for 55.1% and 38.8% of the net energy balance, although this may be biased low due to the inability to accurately capture tree trunk temperature. Although Mazurkiewicz et al., (2008) did not differentiate between radiation terms, they found that net radiation being the largest contributor to melt. At the High sites this trend persists, although the magnitudes change. Within the High-Forest site,

shortwave radiation accounts for the majority of energy received at the snow surface but the annual total is reduced 58 % with net longwave radiation accounting for 42 %. Conversely, at the High-Open site solar radiation accounts for 71 % of the annual total while longwave is reduced to 7 %. The turbulent fluxes account for the remaining 22 %.

The stable atmospheric conditions at all sites, except the High-Open site reduce the turbulent fluxes to consistently insignificant values at the daily time scale with only few days over the course of the study period where these fluxes persist

(Fig. 4). Not surprising then, is the importance of the radiative fluxes on the net energy balance at all sites outside of the High-Open site. Longwave radiation dominates at the Low- and Mid-Forest sites regardless of elevation or year (Fig. S1 – S4). Snowpack melt response to the increased longwave radiation in the forest from lasting events can be substantial. For example, at the Mid-Forest site during an eight-day mid-January period longwave radiation at the snow surface increased 71 w m$^{-2}$ (225 % increase) while snowmelt response was immediate and significant, attributing to a reduction of 32 cm (37 %)

of snowpack depth (Fig. 5). During the same period, longwave radiation increased 56 w m$^{-2}$ (342 % increase) at the Mid-Open site while snowpack was reduced 6 cm (5 %). Throughout WY 2013 longwave radiation inputs are shown to have a strong inverse correlation with snowpack depth at the Mid-Forest site (Fig. 5). This is not the case at the Mid-Open where snowmelt is driven by shortwave radiation with few accumulation season melt events at all, with snowpack settling

attributed to the major snow reduction event in late December. A similar analysis at the High sites shows shortwave radiation driving the snowmelt response to mid-season melt events (Fig. S4). WY 2015 was a historically low year for the Pacific Northwest (Sproles et al., 2017) however, over a four-day period in early January 2015 a large melt event occurred where the High-Forest experienced a 37 % reduction in snow depth and the High-Open snow depth reduced by 50 % (Fig. S4).

Longwave radiation increased 94 % at the Forest site, attributing to 71 % of the total energy budget during the event. Conversely, the Open site longwave radiation increased 366 % yet accounted for only 26 % of the total net energy budget with shortwave radiation at 49 % and the net turbulent flux contributing the rest.

Air temperature is a first order control in longwave radiation calculation and therefore, it is expected that the lower and thus warmer sites experience a larger percentage of net radiation in the form of longwave radiation. Average monthly air

temperatures show that the High-Forest site is 1.9 and 1.8 °C cooler during the winter months (DJF) than the Low- and Mid-Forest sites, respectively (Fig. 6). Colder temperatures reduce the longwave radiation received at the snow surface during the winter months as longwave radiation is non-linearly controlled by air temperature (Eq. 7). The reduced longwave input and lower forest density at the High-Forest site is reflected in the radiation budget where the net longwave energy component is 25 % less than the net longwave energy at the Low and Mid-Forest sites.

Wind speeds at all sites except at the High-Open site are relatively weak and inconsistent resulting in little turbulent mixing. Sustained (annual average) wind speeds at the High-Open site are over five times greater than at any other site with peak daily maximums more than 9 times greater (Fig. 7). At the High-Open site high wind speeds occur frequently while all other sites experience low winds speeds and little variability. Mean winter wind speed for the High-Open site is 3.6 m s$^{-1}$. Mean winter wind speed for the Low- and Mid-Open sites are both 0.7 m s$^{-1}$. The high wind speeds cause instability and

subsequent turbulent mixing resulting in much larger turbulent fluxes at the High-Open site. Conversely, when wind speeds are low minimal, if any, mixing occurs and a decoupling of the snow surface and the atmosphere can persist. Calculation of the Richardson number (Eq. 13) determines the stability of the atmosphere and where values greater than 0.2 this decoupling occurs. Although there is not a consensus of what threshold this critical value should be, we use a threshold of 0.2 (Raleigh et al., 2013). Over the course of the study the $Ri_B$ value within each cover type at the Low and Mid elevation sites and the

High-Forest site exceeds the critical value a majority of day. For example, in WY 2014 the critical value was exceeded 60 % of the time at both the Low sites, 76 % and 71 % at the Mid-Open and Mid-Forest sites, 82 % of the time at the High-Forest site, and only 10 % of the time at the High-Open site.

Forest structure at the Low- and Mid-Forest sites is typified by average crown diameter of 9.4 m and 6.7 m and average leaf area index (LAI) of 2.4 and 2.7, respectively. At the High-Forest site average crown diameter and LAI was

measured as 2.8 m and 1.1 m, respectively. A multi-layered and randomly distributed forest canopy greatly impacts the amount of solar radiation reaching the forest floor through beam attenuation (Campbell, 1986). Forest canopies provide solar shading as the spring progresses and solar angle increases intensifying incoming solar radiation. At the Low- and Mid-Forest sites where canopy interception is high the impact of solar shading becomes less pronounced and snowpack SWE is not preserved late into the spring. With snowfall magnitude essentially the same at the Mid and High elevations we see that

snowpack last much longer into the spring at the High-Forest site when forest shading has a meaningful effect on reducing solar inputs into the snowpack.

## 4 Discussion

In maritime snow zones where winter precipitation is often a mix of rain and snow, multiple mechanisms align to contradict the conventional wisdom that snow is retained longer in forests than in open areas (Link and Marks, 1999; Jost et al. 2007; Musselman et al., 2008). Multi-layered forest cover and a relatively warm forest increase canopy interception efficiency resulting in significant reductions in sub-canopy snow accumulation (Storck et al., 2002). While no significant relationship existed between daily air temperature and $C_{IE}$ within our study ($p > 0.005$), a threshold behavior appears to exist where events under 15 cm seem to be highly correlated with $C_{IE}$. This suggests a non-linear relationship for event-scale canopy interception in dense, relatively warm forests. The slope of trendlines in Fig. (3) show that the dense forests at these Low- and Mid-Forest sites remove a considerable amount of snow from each event significantly reducing subcanopy accumulation. The high snow removal capacities of these forests suggest canopy density is a first order process on snow accumulation.

The highly non-linear relationship between air temperature and incoming longwave radiation formulation is apparent in the net radiation budget analysis. Infrequent cloud-free days and the warm, dense forests of the study area combine to emit a significant amount of longwave radiation to the snow surface (Berris and Harr, 1987; Sicart et al., 2008; Garvelmann et al., 2014). This leads to a positive net snow surface energy balance and mid-winter melt events, most pronounced at the warmer lower elevation sites. With prolonged exposure to longwave emitted by the canopy and the high efficiency of warm forest canopy interception capabilities, low elevation maritime sub-canopy snowpacks are relatively thin and do not persist long enough into the spring season to benefit from forest shading. This creates a radiative paradox where the longwave radiation emitted by dense and relatively warm forest cover exceeds the resulting reduction in shortwave radiation due to forest shading (Sicart et al., 2004; Lawler and Link, 2011; Lundquist et al., 2013). The higher elevation sites experience colder air temperatures, higher wind speeds, and lower forest density, which combine to decrease $C_{IE}$ and the impact of longwave radiation on mid-winter melt events. Furthermore, relatively low ablation rates for the Low- and Mid-Forest sites suggest that forests do provide some radiative shading during the melt season. However, the benefit of solar shading can only be realized if a sufficient snowcover is present. Otherwise, the effects of reduced solar inputs become secondary and it is the accumulation rate, or more precisely, the efficiency of the canopy interception that is the principle control on the date of snow disappearance.

Here, we considered that wind may have an impact on canopy snow unloading and subsequent increases in sub-canopy snow accumulation. While a seasonal mean presents a general view of the wind environment at each Open site, it masks the variability of wind gusts that can drive snow redistribution. Using the 10 min mean wind speeds better depict the wind characteristics that can affect wind redistribution of snow. Pomeroy and Gray (1990) suggest that for wet snow a snow

transport wind threshold of 7 – 10 m s$^{-1}$ measured at 10 m above the ground surface must be exceeded before any redistribution can occur. Using this threshold, the High-Open site measured wind speeds that met or exceeded the lower threshold 9.9 % of the entire record and 14.4 % if we translate measured wind speed to Z = 10 m using a simple wind profile power law. This represents a substantial amount of the snow season and enough to suggest that wind redistribution is possible. More likely is the wind effect on deposition of snowfall. The influence of the forest on the reduction of wind speeds at the High elevation sites can lead to preferential deposition within the forest as the wind speeds attenuate. Once snow is deposited on the ground the wet maritime snow makes it difficult to be redistributed as a result of saltation and suspension. However, the Open site experiences high enough sustained wind speeds to effectively redistribute and transport wet maritime snow from the High-Open site into the adjacent High-Forest site. Although, the magnitude of this redistribution of snow from the Open to the Forest is unknown, it is reasonable to assume that it is not insignificant considering the sustained high winds of the High-Open environment.

The effects of elevation position within a watershed and forest structure on snow persistence can have serious implications within a warming climate. Sproles et al. (2013) documented a 150 m increase in the elevation of the snow line for every 1 °C temperature increase and showed that projected temperature increases of about 2 °C would shift precipitation at 1500 m from snowfall to a rain-snow mix. If that were to occur then forests at that elevation, e.g. the High-Forest site, that now help maintain late spring snowpacks would likely behave more like the lower elevation forests in which snow melt occurs earlier than in the open areas, effectively offsetting any solar shading gains that the forest can provide in the present. Peak SWE and spring runoff would be reduced at these higher elevations. These high elevation forests could lose their dry season "moisture subsidy", suffer increased moisture stress, with wide ranging implications for forest and water resource managers.

**5 Conclusions**

This paper highlights the complex snow-forest process relationships and suggests that forest cover is a principal control on snow persistence due to reduced accumulation from canopy interception and earlier/faster melt due to increased longwave radiation. High density, relatively warm forests have high canopy interception efficiency that controls sub-canopy snowpack evolution and mediates the amount of springtime solar shading of the snowpack. The cooler and less dense High-Forest site has a reduced interception efficiency and acts as a snow deposition reservoir for the nearby windy High-Open site. Net radiation drives the snow surface energy balance with the partitioning between longwave and shortwave a function of forest complexity. Our study demonstrates the sensitivity of Pacific Northwest snowpack development to temperature and forest cover. Nolin and Daly (2006) demonstrated that much of the Oregon Cascade snowpack is at-risk, the ForEST network included, by looking at temperature only. Similarly, Sproles et al., 2013 showed that the lower boundary of the snow zone has little resilience to a warming world. Our paper demonstrates that understanding the snowpack energy budget is key to understanding how forests influence snow accumulation and melt. By quantifying the mechanisms of how

vegetation affects sub-canopy snowpack energy balance the results of this study provide the basis for understanding the sensitivity of maritime snowpacks to a changing climate. As climate continues to warm, we anticipate reduced snow accumulation at elevations where snowfall shifts to a rain-snow mix and amplified sub-canopy melt rates due to longwave radiative heating in warmer forests, thereby reducing overall forest snow retention. Yet, higher elevation colder sites with a less dense forest can mitigate that to some extent by retaining the snowpack longer through lower relative forest longwave emission and lower canopy interception. A key finding within this study is that throughout the study duration, one that saw high inter-annual snowfall variability, a definitive pattern emerged within the energy budget and snowpack dynamics across the network. The energy budget format that we present here goes beyond the temperature only approach while getting at the causal effects and mechanisms of the challenge of vegetation-snowpack interactions for a warming climate.

While these results are focused on the Oregon Cascades, they have broader implications for other relatively warm forested snow environments with elevation gradients, such as parts of the California Sierra Nevada, the Japanese and European Alps, and the Pyrenees (Lundquist et al., 2013). These results will aid in improving parameterizations of snow-forest interactions in physically based snow hydrology models and land surface models. Additionally, as climate change alters regional snow deposition patterns across the western U.S., our findings are applicable to land and water managers, seeking to improve forest snowpack retention, enhance forest health, and improve streamflow forecasting. This study demonstrates the value of plot scale snow-forest process studies for improving our understanding of the forest effects on snowpack evolution. Future work will focus on a multi-scale approach that incorporates remote sensing and snow hydrology modelling to identify forest structure metrics that are well suited to accurately model snow-forest interactions. Such an approach will allow the snow community to quantify the improvement of snow-forest interactions across spatial scales and enhance model prediction for landscape and regional applications.

*Data availability.* The data used in this study are freely available online from the Oregon State University scholars archive https://ir.library.oregonstate.edu/xmlui/handle/1957/59984 (Roth, T. R., & Nolin, A. W., 2016).

## 6 Acknowledgments

This research was made possible by funding provided by National Science Foundation (EAR 1039192) and from a NASA Earth Science Student Fellowship (16-EARTH16F-0426). We thank the Willamette National Forest for providing access permits for the ForEST network. Additional material support was provided by the Western Ecology Division office of the Environmental Protection Agency, with special thanks to Ron Waschmann. We thank the many student interns that assisted in snow surveys and site maintenance.

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

**8 Tables**

**Table 1. Site Forest Characteristics with Associated Standard Deviation for Each Measurment**

| Site | DBH (cm) | Height (m) | Crown Diameter (m) | Forest Density per 10m$^2$ | SVF (%) | Study Duration Average $C_{IE}$ (%) |
|---|---|---|---|---|---|---|
| Low-Forest | 52.1 ± 20.0 | 33.7 ± 10.4 | 9.4 ± 0.8 | 19.4 ± 2.0 | 10.9 | 79 |
| Low-Open | 17.3 ± 4.3 | 8.9 ± 2.6 | 3.7 ± 0.4 | 15.7 ± 5.1 | 68.7 | -- |
| Mid-Forest | 36.5 ± 17.4 | 21.2 ± 10.1 | 6.7 ± 0.8 | 20.7 ± 1.5 | 10.1 | 76 |
| Mid-Open | 19.0 ± 8.3 | 11.8 ± 3.4 | 4.0 ± 0.2 | 15.8 ± 6.7 | 61.6 | -- |
| High-Forest | 21.4 ± 4.1 | 14.2 ± 3.7 | 2.8 ± 0.1 | 19.0 ± 12.8 | 35.1 | 39 |
| High-Open* | 29.4 ± 10.3 | 9.9 ± 3.4 | 0.4 ± 0.6 | 13.1 ± 3.9 | 88.1 | -- |

*Includes fire related standing dead trees

5  **Table 2. Summary Snow Statistics WY 2012 – WY 2014  – Oregon ForEST Network**

| Site | WY2012 | | | WY2013 | | | WY2014 | | |
|---|---|---|---|---|---|---|---|---|---|
| | Peak SWE (cm) | $C_{IE}$ (%) | Ablation Rate (depth cm day$^{-1}$) | Peak SWE (cm) | $C_{IE}$ (%) | Ablation Rate (depth cm day$^{-1}$) | Peak SWE (cm) | $C_{IE}$ (%) | Ablation Rate (depth cm day$^{-1}$) |
| Low-Forest | 23 | 70 | 1.6 | 24 | 75 | 1.9 | 8 | 92 | 0.4 |
| Low-Open | 48 | - | 4.0 | 51 | - | 4.3 | 18 | - | 1.3 |
| Mid-Forest | 45 | 70 | 1.0 | 26 | 75 | 1.3 | 12 | 83 | 1.1 |
| Mid-Open | 89 | - | 3.8 | 73 | - | 2.5 | 38 | - | 4.5 |
| High-Forest | 100 | - | 4.1 | 73 | - | 2.4 | 59 | 39 | 3.1 |
| High-Open | 90 | - | 5.4 | 71 | - | 2.9 | 42 | - | 5.9 |

**9 Figures**

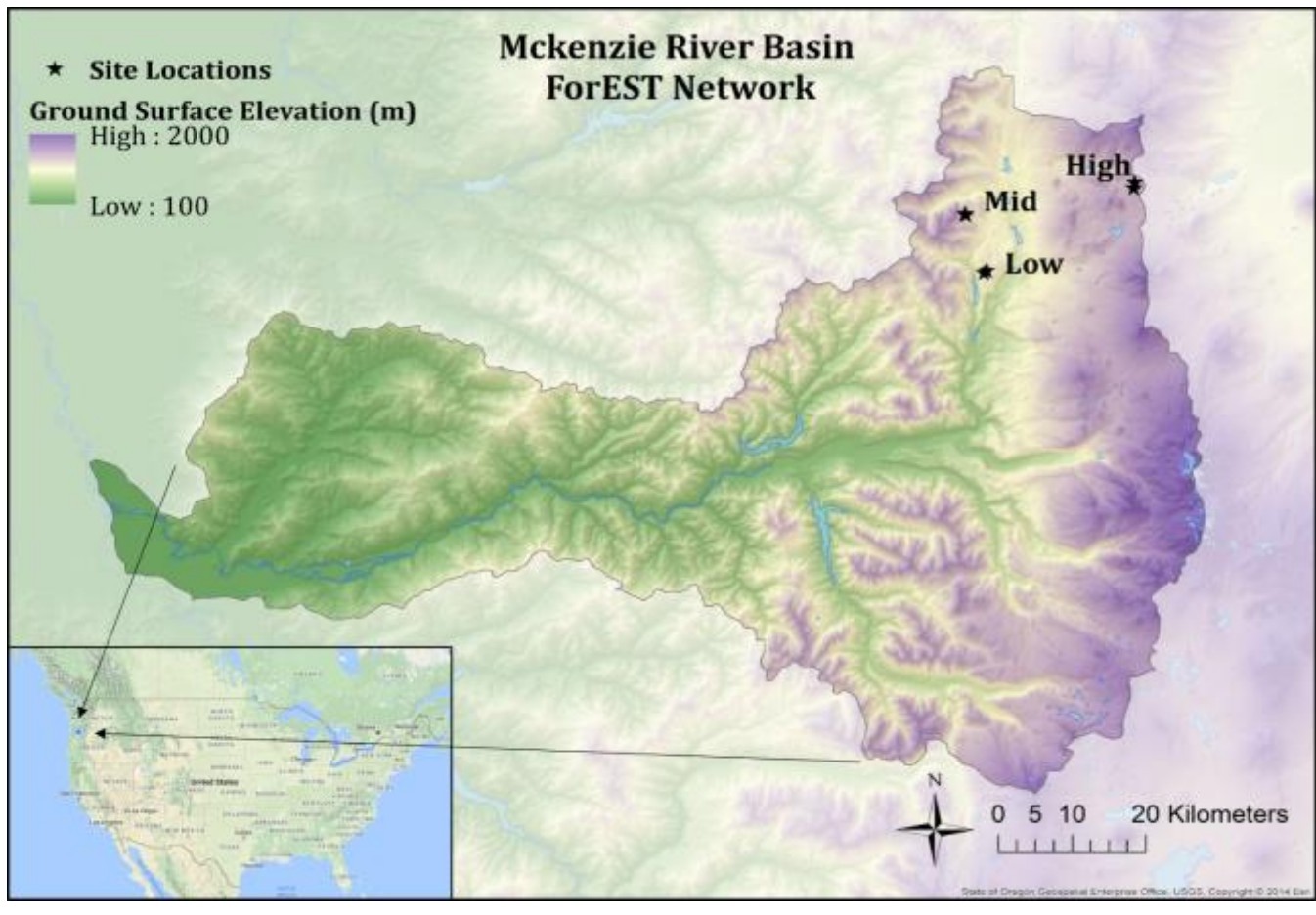

**Figure 1: The Oregon ForEST network sites of the McKenzie River Basin .**

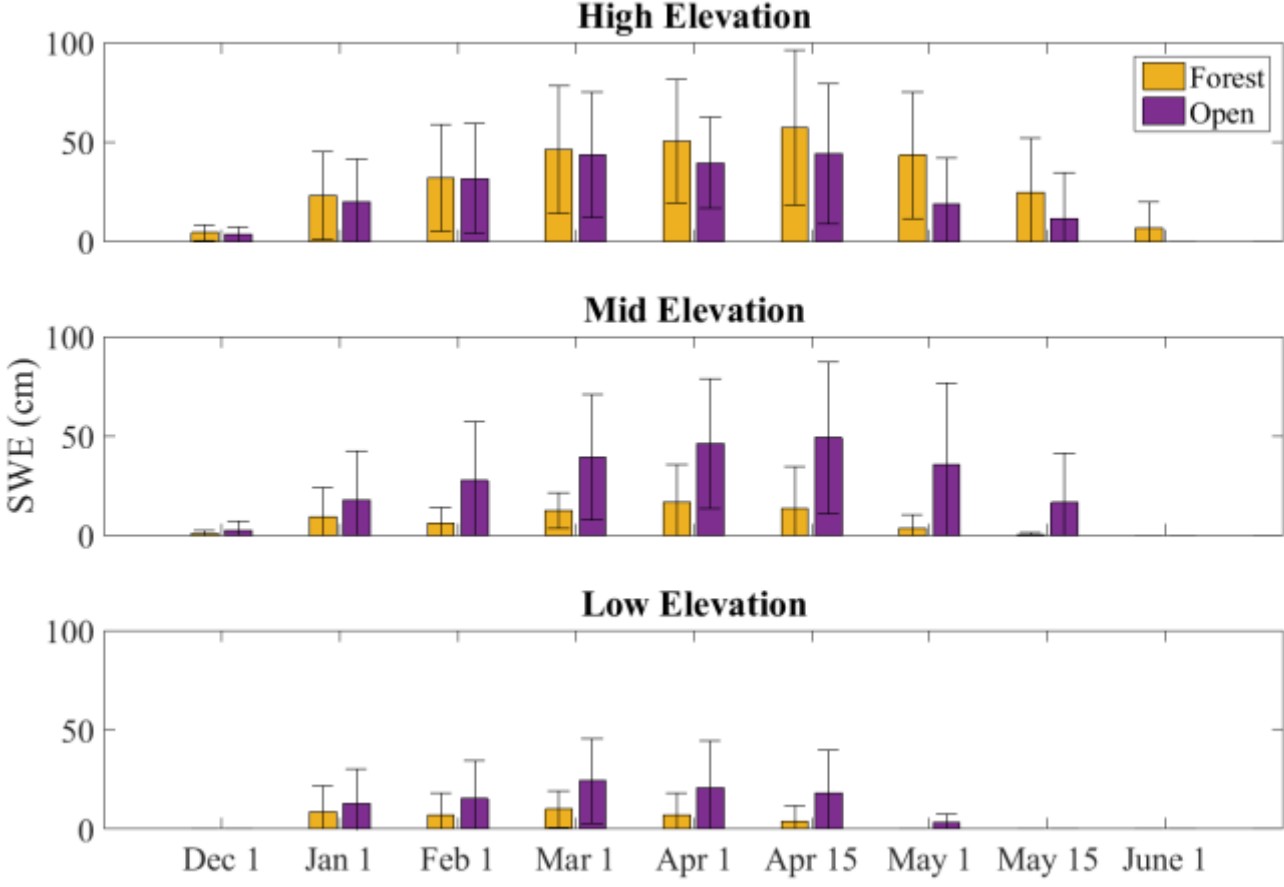

**Figure 2: Average snow water equivalent (SWE) for Open and Forest sites within the ForEST network, WY 2012 – 2014.**

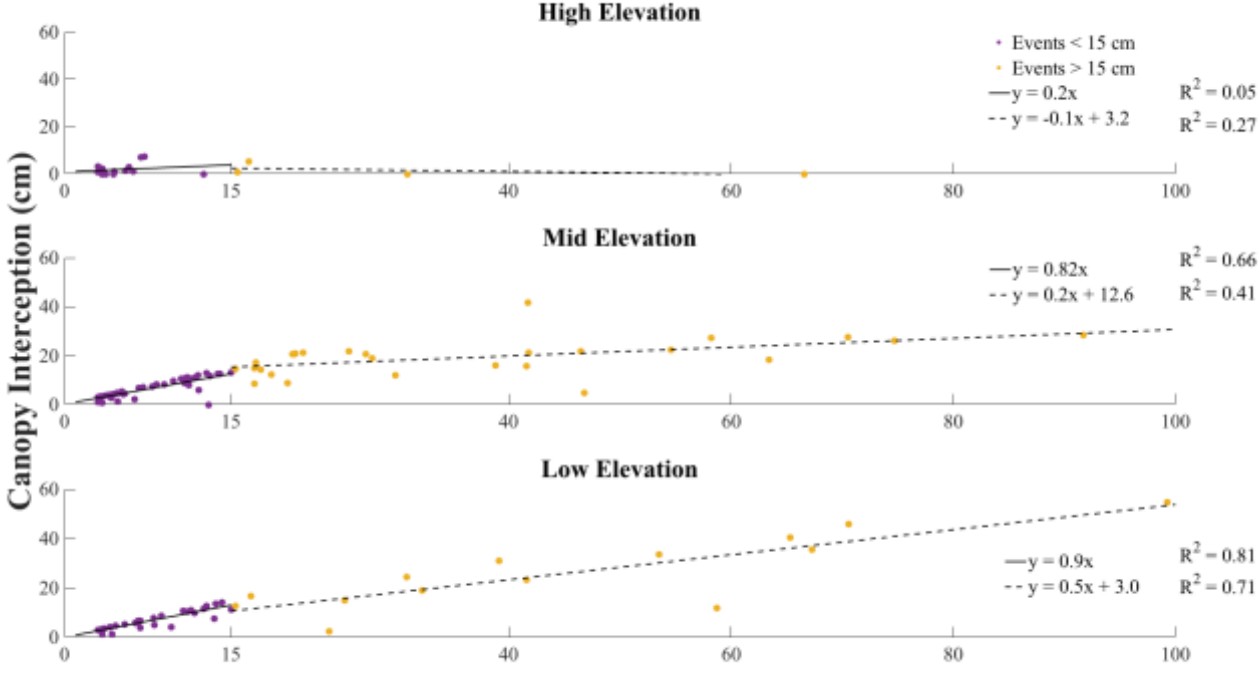

**Figure 3: Canopy interception depth vs. event snowfall within the ForEST network.**

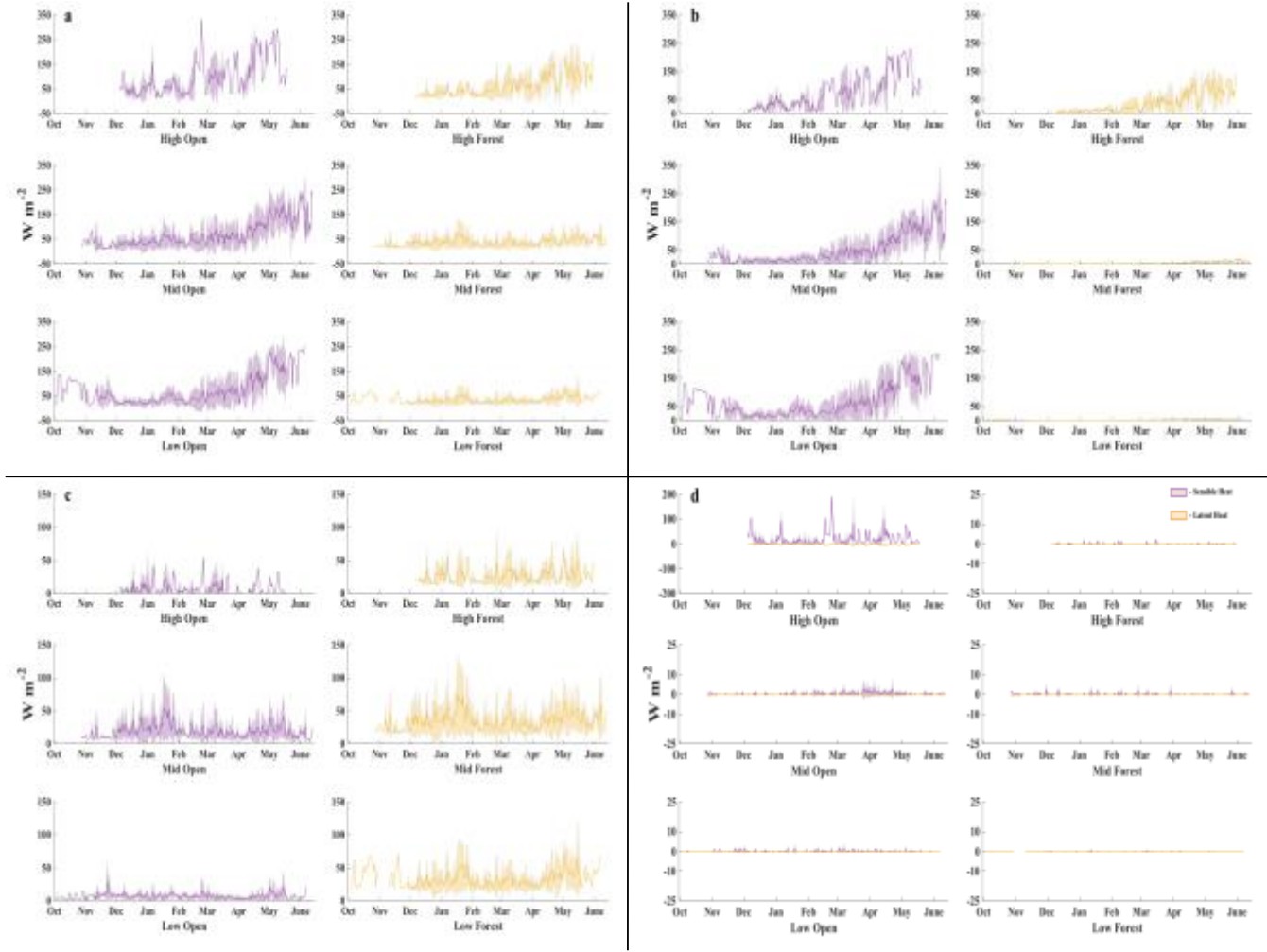

**Figure 4: Calculated daily mean energy balance in W m⁻² (solid line) and the range of values (shaded area) for a) net energy at the snow surface; b) net solar radiation; c) net longwave radiation; and d) net turbulent energy at the snow surface for each site within the ForEST network, WY 2012 – WY 2015.**

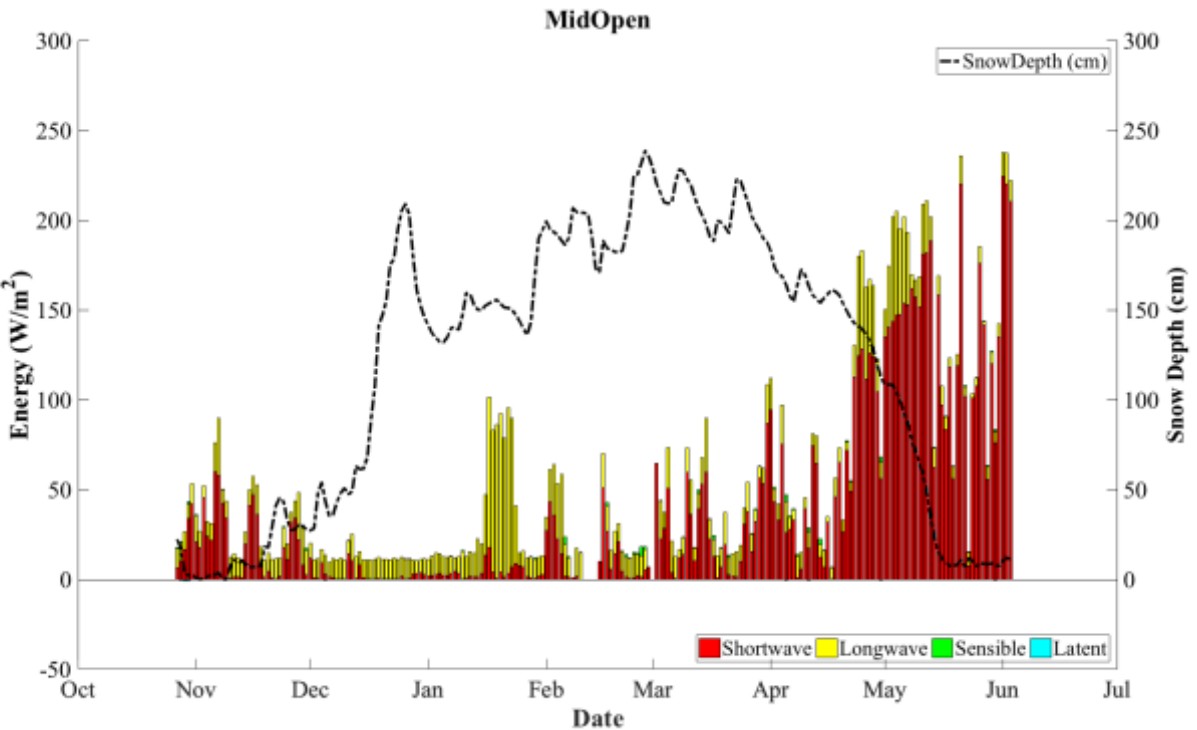

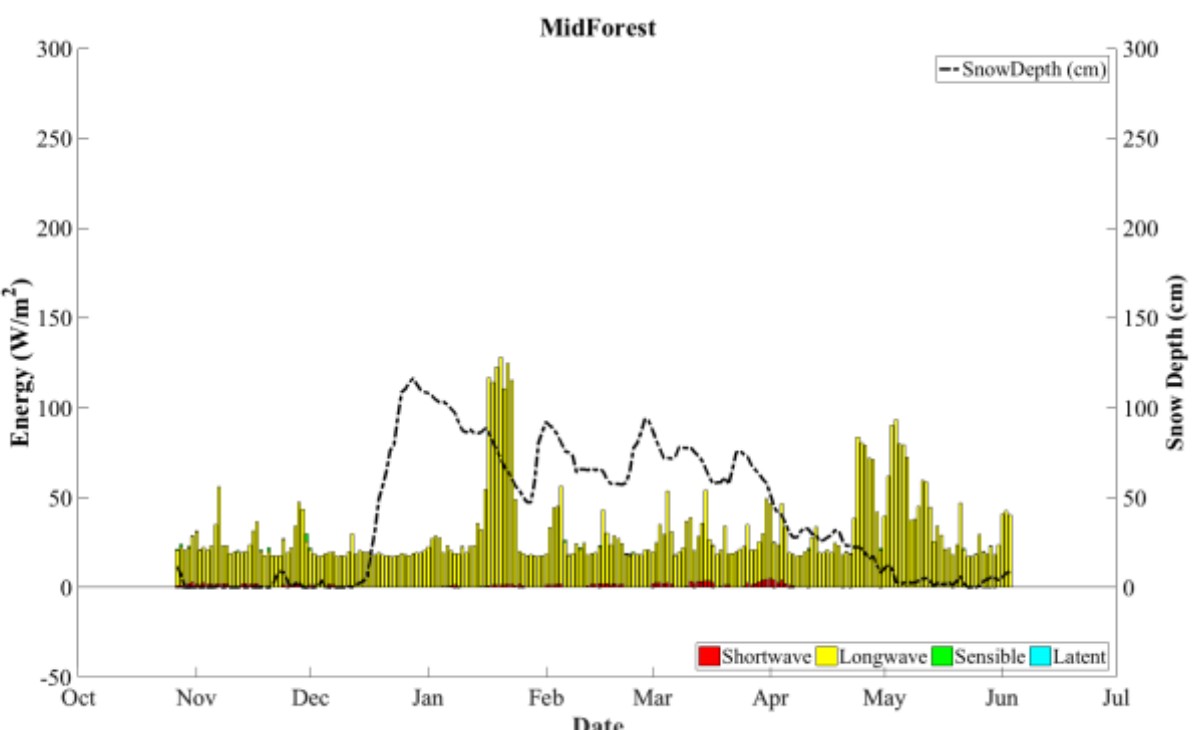

**Figure 5: Calculated daily mean energy balance component magnitudes (bars) and the daily measured snow depth (dashed line) for the Mid-Open (top) and the Mid-Forest (bottom) during WY 2013.**

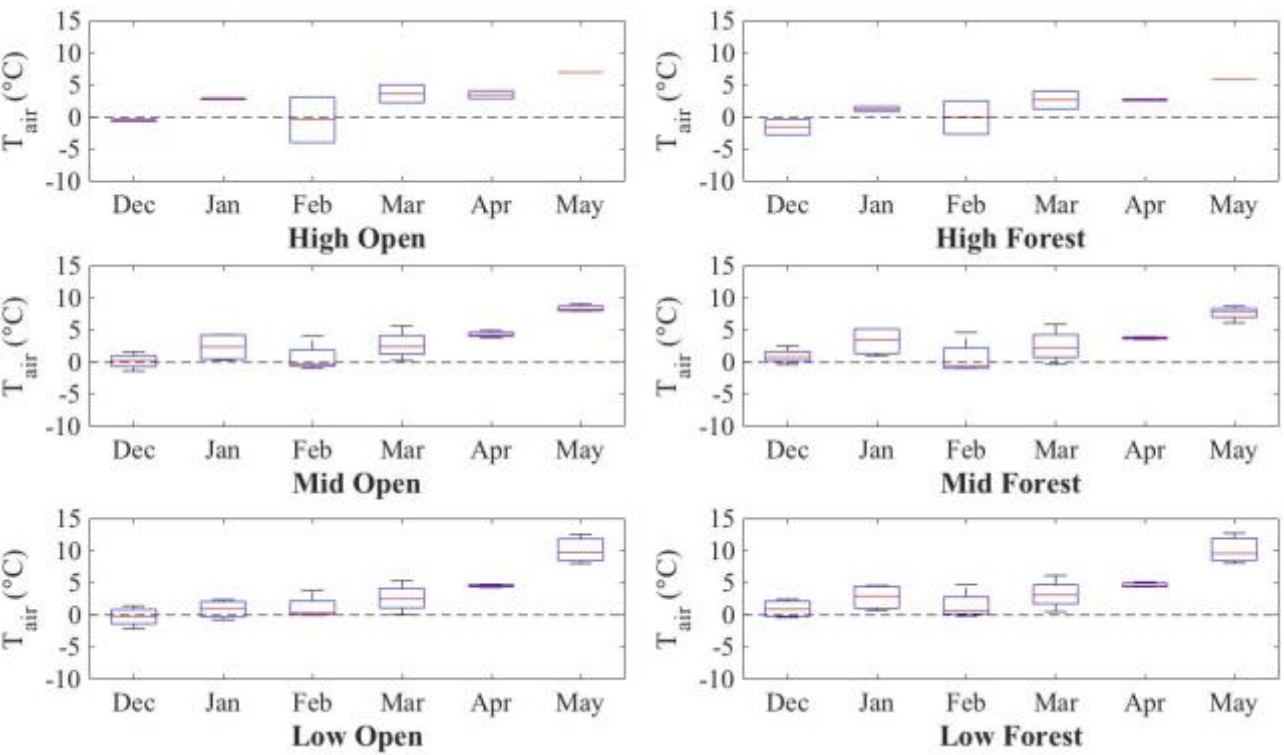

5   **Figure 6: Boxplot of average monthly air temperature for each site within the ForEST network, WY 2012 – WY 2015.**

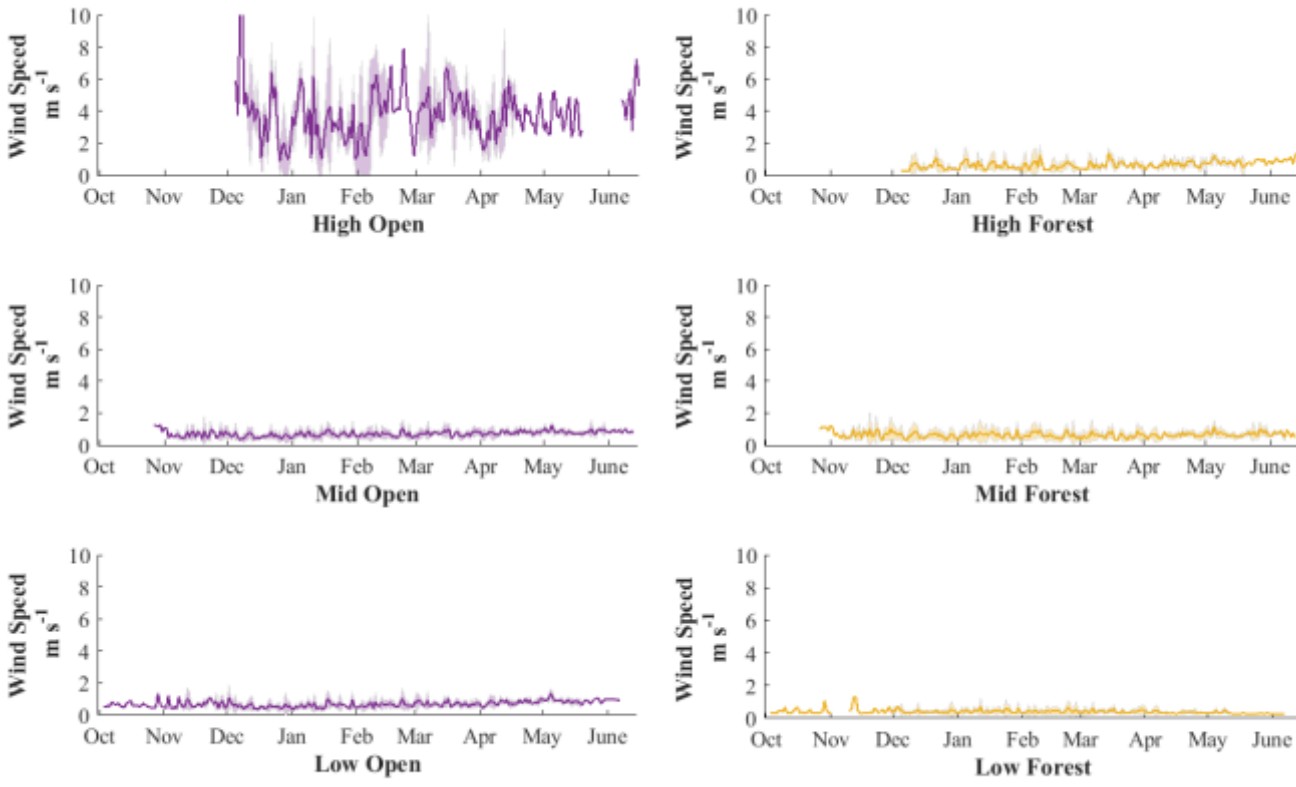

**Figure 7: Daily average wind speed (heavy solid line) and the range of wind speeds (shaded area) for each site within the ForEST network, WY 2012 – WY 2015.**

