# Peer review of "Forest impacts on snow accumulation and ablation across an elevation gradient in a temperate montane environment"

_Hydrology and Earth System Sciences, 2016_

## Referee Comment (RC1) · Anonymous Referee #1 · 2 Dec 2016

**General and specific comments**

This discussion paper deals with an important question (how forests influences net snow precipitation at different altitudes and canopy densities). The site study seems well performed and well documented. Modeling of the surface energy balance (EB) is also performed with several assumptions, and the paper would be improved if the effects of these assumptions were discussed and additional comparisons with other studies were made.

Would, for example, the simulation result be approximately the same if a different formulation of the roughness length, forest floor albedo, canopy temperature, transfer coefficients etc. were used?  Can you e.g. expect the same relationship between forest and air temperature forests with different SVF?  The Pomeroy and Gray (1990) study regarding threshold wind velocity for saltation of snow refer to wind velocity at 10 m, are you using the same measurement height?  Unloading of snow from the branches does not seem to be discussed. The study sites were chosen with regard to slope aspect. What about curvature, which also might influence the accumulation? Snow courses were performed from open areas into forested areas. Were measurements at the forest edges avoided? Otherwise prevailing wind direction might influence the accumulation pattern there.  The open sites had less than 20% canopy cover; did the cover differ between the open sites? If so, how might this have influenced the results?

The paper would also be improved if studies from maritime snowy Japan and Spain were cited (see below) and the early studies by Leaf from the Rocky Mountains who discusses the effect of wind on canopy interception.
I would also appreciate clear information about how events with mixed rain/snow were separated from pure snowfall events.

The paper should be checked to assure consistent use of notations.

**Suggested additional references and technical corrections**

**References**
Possible publications from maritime Japan and from Spain are listed below and other publications by these authors could also be relevant.

López-Moreno, J I., and J. Latron. "Influence of canopy density on snow distribution in a temperate mountain range." *Hydrological Processes* 22.1 (2008): 117-126.

López-Moreno, J. I., et al. "Sensitivity of the snow energy balance to climatic changes: prediction of snowpack in the Pyrenees in the 21st century." *Climate Research* 36.3 (2008): 203-217.

López-Moreno, J. I., S Goyette, and M Beniston. "Impact of climate change on snowpack in the Pyrenees: Horizontal spatial variability and vertical gradients." *Journal of Hydrology* 374.3 (2009): 384-396.

Nakai, Y. et al. "Energy balance above a boreal coniferous forest: a difference in turbulent fluxes between snow-covered and snow-free canopies." *Hydrological Processes* 13.4 (1999): 515-529.

Nakai, Y et a l. "The effect of canopy-snow on the energy balance above a coniferous forest." *Hydrological processes* 13.14-15 (1999): 2371-2382.

Ohta, T., et al. "Characteristics of the heat balance above the canopies of evergreen and deciduous forests during the snowy season." *Hydrological Processes* 13.1415 (1999): 2383-2394.

Lundberg, A, et al. "Snow accumulation in forests from ground and remote-sensing data." *Hydrological Processes* 18.10 (2004): 1941-1955.

Long wave radiation, wind effects and aerodynamic resistance formulation (roughness length) could e.g. be compared with the studies by:

Niemelä, S., P Räisänen, and H Savijärvi. "Comparison of surface radiative flux parameterizations: Part I: Longwave radiation." *Atmospheric Research* 58.1 (2001): 1-18.

Pomeroy, J. W., and K. Dion. "Winter radiation extinction and reflection in a boreal pine canopy: measurements and modelling." Hydrological processes 10.12 (1996): 1591-1608.

Leaf, C.F., "Watershed management in the Rocky Mountain subalpine zone: the status of our knowledge" (1975). *Aspen Bibliography.* Paper 5098. http://digitalcommons.usu.edu/aspen_bib/5098

Beljaars, ACM, and P Viterbo. "The sensitivity of winter evaporation to the formulation of aerodynamic resistance in the ECMWF model." *Boundary-layer meteorology* 71.1-2 (1994): 135-149.

Montesi, J, et al. "Sublimation of intercepted snow within a subalpine forest canopy at two elevations." *Journal of Hydrometeorology* 5.5 (2004): 763-773.

**Technical corrections**

A list of used notations would improve the paper, and the paper should be checked to assure consistent use of notations,

- height above snow surface is sometimes noted with Z and sometimes with z and
- air temperature is sometimes noted with Ta and sometimes with Tair etc.

***Typographical corrections etc..***

P1 L13  remove the word *on*

P2 L15-16. Missing text ?;   range from   to …..

P3 L12-15. Clarify how total amount of snow accumulation can alter time of onset and melt rate! Can it?

P5 L15 Substitute the word *radius* with diameter

P8 L2  Remove *amount of*

P8 L21 add unit for Ce (-)

p12 L 17. By the word complex? Do you mean multilayer?

*Figures*

Fig 2. Add information about used years!

Fig 3. Cm SWE or cm height?

Fig. 5.  The figures would be easier to interpret in the zero degree line was added. This cannot be mean monthly air temperature; it has to be a boxplot of air temperatures.  Is the same type of plot used for all sites and is the same number of years used for all sites?

Figure 6. The shaded areas are not gray. I assume the range is over the day - not over the study duration?

Tables.

Table 1. Is the ablation rate given in snow height or SWE?

Table S2. Remove unnecessary brackets!

---

## Referee Comment (RC2) · Anonymous Referee #2 · 16 Jan 2017

I reviewed the paper "Forest impacts on snow accumulation and ablation across an elevation gradient in a temperate montane environment" by Roth and Nolin for publication in HESS. I found the paper to be a well written explanation of a novel and robust dataset. My comments for major revisions focuses on better linking accumulation and ablation processes with energy budget estimates. Overall, some of the discussion needs to explain to the reader the cumulative importance of the results, particularly in terms of warming temperature and other climatic variability. I outline these as several major and minor comments:

MAJOR COMMENTS:

1. Link energy budget to snowpack observations: The authors' present two very interesting datasets, namely snow physical properties and micrometerology/energy budget. However, not enough effort is made to explain snowpack characteristics with energy budget. Some suggestions are below (some of these are relevant to comment 2 and 3).

a. Do you see correspondence between melt events and energy inputs, both in terms of seasonal and episodic melt. This is difficult to assess because annual snow data is not presented (see comment 3).

b. What about correspondence between melt rates and energy estimates?

c. Can you leverage the different climate (and in particular temperature) to talk about sensitivity of different sites to temperature? See comment 2.

2. Lacking a main take home about "sensitivity": While I think the above comments will help draw out more implications from the results, I would like to see the authors go further in describing the larger implications for 'sensitivity' to drought and warming across these elevations. While the paper's conclusions focus on differences between open and forest canopy, they do not effectively make the case for how the underlying elevation gradient modulates those effects and their corresponding 'sensitivity'. A laymight be some similar to the Nolin and Daly, 2006 classification scheme. I think that the authors should consider how to use the inter-annual variability to explain sensitivity. Consider leaving in 2015 or using it as an example vis a vis the Nolin and Daly classification. How do these differences in snow-vegetation interactions overlay on snow risk for change? What forest position are more likely to see exaggeration of current open/forest snowpack differences? Which are more buffered and why? 3. Better show data in figures/tables: The energy budget time series are useful but difficult to compare. It might be possible to summarize all the sites into a single barplot figure using monthly means. I also think you should show the continuous snow depth time series either as a separate figure or overlayed onto Figure 2. You might consider breaking out Figure 2 by year (see comment 1). Same for Figure 5.

4. Better explain choice/sensitivity to LW algorithm: I would like to see the authors do a better job explaining the longwave radiation models and results to the reader. Can you run a sensitivity analysis with the top 2-3 best models to see if it matters much for your results about the most important heat source being LW. I think making the case a little more strongly for LW will benefit the paper because this is a strong and important finding. Along these lines, add more comparison to previous energy budget and longwave calculations. I would like to know how your net longwave radiation compares to previous measurements in maritime conditions (they seem very high). Here are a coiouple of relevant citations.

- Lapo, K. E., L. M. Hinkelman, E. Sumargo, M. Hughes, and J. D. Lundquist (2017), A critical evaluation of modeled solar irradiance over California for hydrologic and land surface modeling, Journal of Geophysical Research: Atmospheres, n/a–n/a, doi:10.1002/2016JD025527.

- Sicart, J. E., J. W. Pomeroy, R. L. H. Essery, and D. Bewley (2006), Incoming long-wave radiation to melting snow: observations, sensitivity and estimation in Northern environments, Hydrological Processes, 20(17), 3697–3708, doi:10.1002/hyp.6383.

5. Unclear how equation 1 is calculated: It is unclear what time step that interception efficiency is calculated, as the text prior seems to refer to the daily efficiency when snowfall is >3 cm. Figure 3 shows it as a per event ratio. You need to be clear how this is calculated (i.e. Figure 3 does not seem to match equation 1). I like the per event basis.

MINOR COMMENTS:

1. How do estimates of latent heat compare with typical sublimation estimates

2. Add y-axis labels to figure 4

3. May be I missed it but, why did high elevations not intercept snow (e.g. Figure 3)? This is an interesting finding.

---

## Author Comment (AC1) · 14 Feb 2017

Reviewer #1 comments:

1) "Modeling of the surface energy balance (EB) is also performed with several assumptions, and the paper would be improved if the effects of these assumptions were discussed and additional comparisons with other studies were made." The reviewer gives examples of roughness length, forest floor albedo, canopy temperature and transfer coefficients as potential assumptions to be discussed.

Author response: Within our model we deferred to the published literature for any assumed parameter, including those for roughness length [pg. 8, lines: 28-29], forest

floor albedo [pg. 6, line 27], canopy temperature [pg. 8, lines: 5-6], and transfer co-efficients [pg. 8, lines: 23-25]. We recognize that parameter assumptions can have significant effects on the energy budget and their value choices clearly defined, especially in cases were a specific outcome is novel. This paper identifies longwave radiation as significant driver of the forest energy balance and therefore provided analysis on various longwave radiation algorithms. To provide this level of analysis for all parameter choices within the model is beyond the scope of this study.

2) Can we "expect the same relationships between forest and air temperature forests with different SVF?"

Author response: The SVF for each site was measured using hemispherical photographs and Gap Light Analysis software [pg. 5, lines: 16-20]. Therefore, SVF is not a "tunable" parameter but rather a physical forest characteristic specific to each site within our network. How this specifically changes any relationship is inherent within the longwave radiation calculation, e.g. Eq. 7 [pg. 8, line: 1].

3) "The Pomeroy and Gray (1990) study regarding threshold wind velocity for saltation of snow refer to wind velocity at 10m, are you using the same measurement height?"

Author response: No. Our meteorological stations are ∼2m above the snow-free ground surface and during the winter are < 2m above the surface due to snow accumulation. The High-Open site exceeds cited 7-10 m s-1 a full 9.9% [ pg. 13, line: 11] of the entire record and if measured at 10m this exceedance percentile would increase due to the logarithmic relationship between wind speed and height above ground.

4) "Unloading of snow from the branches does not seem to be discussed."

Author response: We note that snow unloading from the canopy is likely with wind the likely main influence [pg. 13, line: 6]. Snow unloading from the canopy was not directly measured in this study and therefore any substantial discussion would be qualitative and based on the author's experience within these forests, not rooted in quantitative

analysis.

5) "The study sites were chosen with regard to slope aspect. What about curvature, which also might influence the accumulation?"

Author response: The study sites were chosen not based on slope and aspect but rather through a binary regression tree analysis that classified the basin in terms of basin peak SWE. This binary regression tree analysis identified elevation, vegetation type, and vegetation density as significant predictor variables [pg. 5, lines: 5-11]. While slope and aspect were not used to specifically identify potential sites we did try to choose sites with minimal slope and similar aspect. Curvature was not considered. A full description of the site selection methodology is outlined in Gleason et al. 2017 (cited [pg. 5, line: 7]).

6) "Snow courses were performed from open areas into forested areas. Were measurements at the forest edges avoided? Otherwise prevailing wind direction might influence the accumulation pattern there."

Author response: In designing our snow courses we took into account that forest edges can influence SWE measurements and therefore maintained a ~50m buffer from a SWE measurement and a forest edge. This will be clarified within the text.

7) "The open sites had less than 20% canopy cover; did the cover differ between the open sites? If so, how might this have influenced results?"

Author response: Canopy cover between open sites did differ. Table 2 [pg. 18] shows forest characteristics of each site, including DBH, height, crown diameter, forest density, and SVF. Forest density is a principle result we present in the paper, specifically how forest density significantly influences the radiation budget and snow accumulation processes [pg. 11, lines: 22-24; pg. 12, lines: 20-32; pg. 13, lines: 1-5].

8) "This paper would also be improved if studies from maritime snowy Japan and Spain were cited and the earlier studies by Leaf from the Rocky Mountains who discusses

the effect of wind on canopy interception."

Author response: The author is familiar with the works of Lopez-Moreno and agrees that this paper would benefit from inclusion of citing his important work. The revised manuscript will reflect this. The works of Nakai et al., (1999 and 1999) and Ohta et al., (1999) are meaningful works that help to provide clarity on the complex interplay of forests and snow processes. However, these studies are constrained to the energy balance above the canopy whereas this paper focuses on the sub-canopy processes primarily.

9) The reviewer "would also appreciate clear information about how events with mixed rain/snow were separated from pure snowfall events."

Author response: This study was focused on forest impacts on snow accumulation and ablation processes from the rain/snow transition line into the seasonal snow zone. As such, this study was implicitly designed to understand the impacts of temperature (elevation) on sub canopy snow processes. The separation of mixed phase (or rain on snow events) from pure snowfall events was not a focus of this study and therefore not considered. The only criteria for snowfall events was a >3cm daily snowfall accumulation depth [pg. 5, lines: 28-31].

10) Technical and typographic corrections cited by the reviewer will be addressed and corrected as requested.
* * *

---

## Author Comment (AC2) · 14 Feb 2017

Reviewer #2 comments:

1) "Link energy budget to snowpack observations: The authors' present two very interesting datasets, namely snow physical properties and micrometeorology/energy budget. However not enough effort is made to explain snowpack characteristics with energy budget." a. "Do you see correspondence between melt events and energy inputs, both in terms of seasonal and episodic melt? This is difficult to assess because annual snow data is not presented."

Author response: We regret not including annual snow data into the paper as the inclu-

sion of these data would provide necessary clarity to the corresponding link between energy budget calculations and measured snow accumulation/melt. We do see temporal links between episodic events and calculated energy budget magnitudes. The revision of this paper will have an updated figure that both includes daily snow depth dynamics and daily energy budget component magnitudes. We are confident that the new figures will address this comment and comment #3 sufficiently.

b. "What about correspondence between melt rates and energy estimates?"

Author response: Comment 1a and 1b are linked and we will address these both by including new, clearer energy budget figures. Additionally, within the results section we will provide clarity as to the level of correspondence between melt rates and energy estimates.

c. "Can you leverage the different climate (and in particular temperature) to talk about sensitivity of different sites to temperature?"

Author response: We will discuss this in our response to comment #2 (below) as this sub-comment pertains to our response directly.

2) "Lacking a main take home about "sensitivity": While I think the above comments will help draw out more implications from the results, I would like to see the authors go further in describing the larger implications for 'sensitivity' to drought and warming across these elevations. While the paper's conclusions focus on differences between open and forest canopy, they do not effectively make the case for how the underlying elevation gradient modulate those effects and their corresponding scheme. A lay might be some similar to the Nolin and Daly, 2006 classification scheme. I think that the authors should consider how to use the inter-annual variability to explain sensitivity. Consider leaving in 2015 or using it as an example vis a vis the Nolin and Daly classification. How do these differences in snow-vegetation interactions overlay on snow risk for change? What forest position are more likely to see exaggeration of current open/forest snowpack differences? Which are more buffered and why?
Author response: The question of the sensitivity of the snowpack to changes in temperature is extremely important for this study region and we welcome the chance to discuss this here. Nolin and Daly (2006) demonstrated that much of the Oregon Cascade snowpack is at-risk, the ForEST network sites included, by looking at temperature only. Similarly, Sproles et al., 2012 showed that the lower boundary of the snow zone has little resilience to a warming world. Both show that there is essentially no resiliency to warming temperatures. Our paper goes further and shows that the understanding the energy budget of the snow surface is most important because we look at the influences of vegetation on snowpack. By showing the mechanisms of how vegetation effects the sub-canopy snowpack energy balance this study provides the basis for truly understanding the sensitivity these temperate forest snowpacks have to warming temperatures. As the climate warms, not only will the frequency of precipitation falling as rain increase longwave radiation will amplify melt as the forest warms. The forests will exacerbate melt where we see dense mature forests. Yet the higher, cooler site with a less dense forest can mitigate that to some extent by retaining the snowpack longer through lower relative forest longwave emission and lower canopy interception. As to the comment about inter-annual variability, the network experienced high inter-annual variability and a key finding that was that throughout a definitive pattern emerged within the energy budget and snowpack dynamics across the network. Sproles et al., (2017) shows that WY2014 can be used as an analog for "average" winter snowpacks in a warming climate whereas the extremely warm WY2015 will continue to be considered a low probability winter in the future. The energy budget format that we present here goes beyond the temperature only approach while getting at the casual effects and mechanisms of the challenge of vegetation for a warming climate.

3) "Better show data in figures/tables: The energy budget time series are useful but difficult to compare. It might be possible to summarize all the sites into a single barplot figure using monthly means. I also think you should show the continuous snow depth time series either as a separate figure or overlayed onto Figure 2. You might consider breaking out Figure 2 by year (see comment 1). Same for Figure 5."

Author response: We intend to revamp these two figures to better depict the response of snow to energy budget calculations (see response #1).

4) "Better explain choice/sensitivity to LW algorithm: I would like to see the authors do a better job explaining the longwave radiation models and results to the reader. Can you run a sensitivity analysis with the top 2-3 best models to see if it matters much for your results about the most important heat source being LW. I think making the case a little more strongly for LW will benefit the paper because this is a strong and important finding. Along these lines, add more comparison to previous energy budget and longwave calculations. I would like to know how your net longwave radiation compares to previous measurements in maritime conditions (they seem very high)."

Author response: Understanding that our longwave radiation result was a strong and important finding we share this reviewer's concern as to the calculation of longwave radiation and the sensitivity to its formulation and given the infrequency of cloud-free days, the cloud correction algorithm. With that in mind we ran a comparative analysis on a multitude of longwave calculation formulations much in the same as Flerchinger et al., (2009) and reported the root-mean squared errors of each algorithm [Table S2 and Table S3]. A full description of the study is given [pg. 7, lines 20-32]. Additionally, the authors agree in the value of the reviewer's suggestion as to how our longwave calculations findings compare to previous measurements in maritime conditions. Addition of this will benefit the paper and provide necessary context.

5) "Unclear how equation 1 is calculated: It is unclear what time step that interception efficiency is calculated, as the text prior seems to refer to the daily efficiency when snowfall is >3 cm. Figure 3 shows it as a per event ratio. You need to be clear how this is calculated (i.e. Figure 3 does not seem to match equation 1). I like the per event basis."

Author response: Eq. 1 is calculated on a per event basis. This ambiguity will be clarified in the text to reflect both the per event basis of Eq. 1 and how an event is

constrained.

Minor comments: 1) "How do estimates of latent heat compare with typical sublimation estimates?"

Author response: We did not make any comparisons with other published sublimation results and will include some discussion to address this comment. Adding some comparative analysis will give the reader a better understanding of the underlying processes of our study area.

2) "Add y-axis labels to figure 4."

Author response: We will fix this minor issue with Figure 4.

3) "May be I missed it but, why did high elevations not intercept snow (e.g. Figure 3)? This is an interesting finding."

Author response: We suggest that the low interception capacities at the High-Forest site is based on canopy density and note that this removal inefficiency is a first order process on seasonal snow accumulation [pg. 12, lines: 23-24].

―――――――――――――――――――

---

## Author Response (AR1)

**Referee comments responses**

Comment:

5      "Modeling of the surface energy balance (EB) is also performed with several assumptions, and the paper would be improved if the effects of these assumptions were discussed and additional comparisons with other studies were made." The reviewer gives examples of roughness length, forest floor albedo, canopy temperature and transfer coefficients as potential assumptions to be discussed.

10     **Author response**: Within our model we deferred to the published literature for any assumed parameter, including the referee suggested parameters of roughness length [pg. 8, lines: 29-30; pg. 9, lines: 1-2], forest floor albedo [pg. 6, lines 28-29], canopy temperature [pg. 8, lines: 3-6], and transfer coefficients [pg. 8, lines: 23-25]. We recognize that parameter assumptions can have significant effects on the energy budget and their value choices clearly defined, especially in cases were a specific outcome is novel. This paper identifies longwave radiation as significant

15     driver of the forest energy balance and therefore provided an in-depth analysis on various longwave radiation algorithms. To provide this level of analysis for all parameter choices within the model is beyond the scope of this study.

Comment:

20     Can we "expect the same relationships between forest and air temperature forests with different SVF?"

       **Author response**: The SVF for each site was measured using hemispherical photographs and Gap Light Analysis software [pg. 5, lines: 19-23]. Therefore, SVF is not a "tunable" parameter but rather a measured physical forest characteristic specific to each site within our network. How this specifically changes any relationship is inherent

25     within the longwave radiation calculation, e.g. Eq. 7 [pg. 8, line: 1].

Comment:

       "The Pomeroy and Gray (1990) study regarding threshold wind velocity for saltation of snow refer to wind velocity at 10m, are you using the same measurement height?"

       **Author response**: We have clarified the height of our wind speed measurements within the text [pg. 8, lines: 27-28]. Our meteorological stations are 3 m above the snow-free ground surface for the Low and Mid sites, and 4.5 m at the High sites. The High-Open site exceeds the Pomeroy & Gray (1990) cited 7-10 m s$^{-1}$ a full 9.9% [pg. 13, line:

31-34] of the entire record and if we recalculated measured wind speeds using a simple wind profile power law to a 10 m reference height this exceedance percentile would increase to 14.4%.

Comment:

"Unloading of snow from the branches does not seem to be discussed."

**Author response**: We note that snow unloading from the canopy occurs due to wind [pg. 13, line: 27-28]. Snow unloading from the canopy was not directly measured in this study and therefore any substantial discussion would be qualitative and based on the author's experience within these forests, not rooted in quantitative analysis.

Comment:

"The study sites were chosen with regard to slope aspect. What about curvature, which also might influence the accumulation?"

**Author response**: The study sites were chosen not based on slope and aspect but rather through a binary regression tree analysis that classified the basin in terms of basin peak SWE. This binary regression tree analysis identified elevation, vegetation type, and vegetation density as significant predictor variables [pg. 5, lines: 7-14]. While slope and aspect were not used to specifically identify potential sites we did try to choose sites with minimal slope and similar aspect. Curvature was not considered. We clarified the text to ease readability and address this referee comment [pg. 5, line: 11]. A full description of the site selection methodology is outlined in Gleason et al. 2017 (cited [pg. 5, line: 10]).

Comment:

"Snow courses were performed from open areas into forested areas. Were measurements at the forest edges avoided? Otherwise prevailing wind direction might influence the accumulation pattern there."

**Author response**: In designing our snow courses we took into account that forest edges can influence SWE measurements and therefore maintained a ~50 m buffer from a SWE measurement and a forest edge. This is clarified within the text [pg. 4, lines: 31-32].

Comment:

"The open sites had less than 20% canopy cover; did the cover differ between the open sites? If so, how might this have influenced results?"

**Author response**: Canopy cover between open sites did differ. Table 2 [pg. 20] shows forest characteristics of each site, including DBH, height, crown diameter, forest density, and SVF. Forest density effects on sub-canopy snow dynamics is a principle result we present in the paper, specifically how forest density significantly influences the radiation budget and snow accumulation processes [pg. 12, lines: 26-30; pg. 13, lines: 4-26].

Comment:

"This paper would also be improved if studies from maritime snowy Japan and Spain were cited and the earlier studies by Leaf from the Rocky Mountains who discusses the effect of wind on canopy interception."

10     **Author response**: Thank you for these references. We are familiar with the papers of Lopez-Moreno and have now included them as citations [pg. 2, lines: 3 & 5]. The works of Nakai et al., (1999 and 1999) and Ohta et al., (1999) are meaningful however, these studies are constrained to the energy balance above the canopy whereas this paper focuses on the sub-canopy processes.

15     Comment:

The reviewer "would also appreciate clear information about how events with mixed rain/snow were separated from pure snowfall events."

    **Author response**: We have clarified the text to address this comment citing that the only criteria for snowfall
20     events was a >3 cm daily snowfall accumulation depth regardless of phase [pg. 5, lines: 28-31]. This study was focused on forest impacts on snow accumulation and ablation processes from the rain/snow transition line into the seasonal snow zone. As such, this study was implicitly designed to understand the impacts of temperature (elevation) on sub canopy snow processes. The separation of mixed phase (or rain on snow events) from pure snowfall events was not a focus of this study and therefore not considered.

Comment:

The comment: "P3 L12-15. Clarify how total amount of snow accumulation can alter time of onset and melt rate! Can it?"

30     **Author response**: We have clarified our intent with the statement in question within the text to convey that *all* of the forest effects on the sub-canopy energy and mass balance can modify the onset and rate of snowmelt. We regret not making this clearer in the original text and have clarified the text [pg. 3, line: 14].

Technical and typographic corrections:

Within text:

- Corrected Z height notation throughout the manuscript to be consistent.
- Corrected $T_a$ to $T_{air}$ throughout the manuscript.
- Addressed the identified typographical errors cited by referee #1 throughout the manuscript, as noted in the marked up text, with the exception of the following:
  - On page 2 lines 16-17 there is no missing text as the sentence states: "Forest disturbance can have significant impacts on snow processes whose effects can **range from** immediate (Boon, 2009) **to** decadal (Lyon et al., 2008; Gleason and Nolin, 2016)."
  - Crown radius was a measured forest characteristic and thus to change the word radius to diameter within the text alters this parameter and the values collected within the field.
  - The bulk transfer coefficient ($C_e$) is a dimensionless variable and therefore the unit is (-).

Figures:

- Figure 2 has been updated to include information about years used.
- Figure 3 has cm height as units and has been updated to clarify that point.
- Figure 5 (now 6) we included a dashed line at 0 °C.
- Figures 4 and 6 (now 7) we clarified the shaded area meaning and included study years into the caption.

Tables:

- Updated Table 1 to reflect ablation rates are in depth cm day$^{-1}$ and not cm SWE.
- Updated Table S2, removed unnecessary brackets.

Comment:

"Link energy budget to snowpack observations: The authors' present two very interesting datasets, namely snow physical properties and micrometeorology/energy budget. However not enough effort is made to explain snowpack

5       characteristics with energy budget. Do you see correspondence between melt events and energy inputs, both in terms of seasonal and episodic melt? This is difficult to assess because annual snow data is not presented."

**Author response**: Thank you for this comment. We regret not including annual snow data into the original manuscript paper as the inclusion of these data provides necessary clarity to the corresponding link between energy

10      budget calculations and measured snow accumulation/melt. We do see temporal links between episodic events and calculated energy budget magnitudes. We have included a supplement analysis that presents melt events within an energy budget context, which will provide the reader with adequate evidence supporting our original findings of longwave radiation dominance within the forested environments. We have updated the manuscript text [pg. 11-12, lines: 24-34 & 1-5] and included an additional figure [Fig. 5] to address this comment.

Comment:

"What about correspondence between melt rates and energy estimates?"

**Author response**: We addressed this in the above comment while including additional supplemental energy figures

20      that are coupled with snow depth measurements and shown for each year within the study duration [Fig. S1 – S4].

Comment:

"Can you leverage the different climate (and in particular temperature) to talk about sensitivity of different sites to temperature?"

**Author response**: We appreciate this referee's comment and have updated the text to included discussion on the sites sensitivity to warming [pg. 14-15, lines: 26-32 & 1-5]. Along with the additional figures presented in response to comments 1 and 2 we feel this manuscript will be better off.

30      Comment:

"Lacking a main take home about "sensitivity": While I think the above comments will help draw out more implications from the results, I would like to see the authors go further in describing the larger implications for 'sensitivity' to drought and warming across these elevations. While the paper's conclusions focus on differences between open and forest canopy, they do not effectively make the case for how the underlying elevation gradient

modulate those effects and their corresponding scheme. A lay might be some similar to the Nolin and Daly, 2006 classification scheme. I think that the authors should consider how to use the inter-annual variability to explain sensitivity. Consider leaving in 2015 or using it as an example vis a vis the Nolin and Daly classification. How do these differences in snow-vegetation interactions overlay on snow risk for change? What forest position are more likely to see exaggeration of current open/forest snowpack differences? Which are more buffered and why?

**Author response**: The question of the sensitivity of the snowpack to changes in temperature is extremely important for this study region and by including a more in depth discussion as to the snowpack sensitivity to warming temperatures and inter-annual variability will benefit this paper. We've updated the text to reflect this [pg. 14-15, lines: 26-32 & 1-5].

Comment:

"Better show data in figures/tables: The energy budget time series are useful but difficult to compare. It might be possible to summarize all the sites into a single barplot figure using monthly means. I also think you should show the continuous snow depth time series either as a separate figure or overlayed onto Figure 2. You might consider breaking out Figure 2 by year (see comment 1). Same for Figure 5."

**Author response**: We have included new figures to the manuscript that depict both daily energy balance components overlain by snow depth [Fig.5 and Fig. S1 – S4]. These new figures should help the reader to better interpret causal relationships between energy components and snowpack dynamics.

Comment:

"Better explain choice/sensitivity to LW algorithm: I would like to see the authors do a better job explaining the longwave radiation models and results to the reader. Can you run a sensitivity analysis with the top 2-3 best models to see if it matters much for your results about the most important heat source being LW. I think making the case a little more strongly for LW will benefit the paper because this is a strong and important finding. Along these lines, add more comparison to previous energy budget and longwave calculations. I would like to know how your net longwave radiation compares to previous measurements in maritime conditions (they seem very high)."

**Author response**: Understanding that our longwave radiation result was a strong and important finding we share this reviewer's concern as to the calculation of longwave radiation and the sensitivity to its formulation and given the infrequency of cloud-free days, the cloud correction algorithm. With that in mind we ran a comparative analysis on a multitude of longwave calculation formulations much in the same as Flerchinger et al., (2009) and reported the root-mean squared errors of each algorithm [Table S2 and Table S3]. A full description of the study is given [pg. 7,

lines 20-31].

Additionally, the authors agree in the value of the reviewer's suggestion as to how our longwave calculations findings compare to previous measurements in maritime conditions. We have expounded on how our longwave radiation findings relate to other similar maritime studies [pg. 11, lines: 10-16].

Comment:

"Unclear how equation 1 is calculated: It is unclear what time step that interception efficiency is calculated, as the text prior seems to refer to the daily efficiency when snowfall is >3 cm. Figure 3 shows it as a per event ratio. You need to be clear how this is calculated (i.e. Figure 3 does not seem to match equation 1). I like the per event basis."

**Author response**: Eq. 1 is calculated on a per event basis. Fig. 3 shows event snowfall vs. canopy interception. $C_{IE}$ is the efficiency of a forest to intercept snow and not the amount of snow the forest intercepted. Clarifying text can be found on [pg. 10, lines: 20-22].

Minor comments:

Comment:

"How do estimates of latent heat compare with typical sublimation estimates?"

**Author response**: We did not analyze sublimation specifically within this study, instead focused on ablation only. Latent heat was only a factor in ablation at one of six sites within the ForEST network and therefore sublimation was not considered individually.

Comment:

"Add y-axis labels to figure 4."

**Author response**: Fig. 4 (now Fig. 5) figure caption was updated to include clarification as to what the y-axis is. The y-axis itself already had an axis label.

Comment:

"May be I missed it but, why did high elevations not intercept snow (e.g. Figure 3)? This is an interesting finding."

**Author response**: We show that the low interception capacities at the High-Forest site is based on canopy density and note that this removal inefficiency is a first order process on seasonal snow accumulation [pg. 13, lines: 8-11].

Results do show that there is an apparent threshold behavior with event size and canopy interception efficiency [pg. 10, lines: 26-30].

[revised manuscript text omitted]

---

## Author Response (AR2)

SUMMARY AND RECOMMENDATION

This manuscript describes the effect of forest cover on snow with elevation in a Maritime basin in Oregon, USA. The ForEST network is described in a previous HESS paper by the authors (Gleason et al., 2017), and includes paired forest-open staations at low, middle, and high elevations. The study examines SWE from snow surveys (with a federal sampler) made across multiple years at the six sites, showing greater springtime SWE and later snow persistence in open areas (vs. forests) at low and mid elevations, but in forested areas at higher elevations. They examine canopy interception efficiency at the different elevations, general finding greater efficiency and the lower and middle elevations vs. the high elevation. They attempt to connect the changes in snowpack to the daily energy balance (forest vs. open) using observations and estimates (e.g., longwave, sensible, latent), finding that the radiative fluxes dominate at all sites, but the importance of longwave vs. shortwave radiation is the result of forest canopy presence/absence. The role of wind speeds in redistributing the snow is discussed, with a greater perceived impact at the high elevations only (which may redistribute snow from open areas to the forest).

This paper makes a valuable contribution to the literature on a topic that has attracted much interest recently. The unique observations from the ForEST network are summarized well and contribute to recent hypotheses posed on snow persistence with respect to forest and temperature/climate. I have found some potential issues in the energy balance estimation that require additional attention, but I do not expect the conclusions to change after making these corrections. I recommend publishing this paper in HESS after all these aspects are addressed. I have also suggested a number of technical edits for clarity, etc.

GENERAL COMMENTS

1. The snow surface temperature is approximated here with the dewpoint temperature (section 2.4). In other words, the vapor pressure at the snow surface ($e\_0$) is assumed equal to the vapor pressure of air at measurement height ($e\_a$). As discussed in Raleigh et al. (2013), one limitation of this simplification is that it results in an elimination of the vapor pressure gradient, thereby muting latent energy exchange (i.e., $e\_a$ should equal $e\_0$ in equation 11, so latent flux should be zero). However, it appears this is not the case in Figure 4, as the latent energy is non-zero. This has a bearing on the relative contribution of turbulent vs. radiative fluxes (section 3.3). Can you please clarify this discrepancy? Raleigh et al. (2013) state that in their analysis of the potential of using $T_{dew}$ as an assumption for the rarely measured $T_{snow}$ there are two caveats. The first being that, by using a daily average $T_{dew}$ to estimate $T_{snow}$ , the diurnal variation typically exhibited by the snow surface temperature is masked. And secondly, and to your point, is that they assumed no vapor pressure gradient as $T_{snow}$ is equal to $T_{dew}$ and therefore $e\_a - e\_o = 0$. As such we re-examined our code and found an error in our latent heat calculation and have updated Figures 4, 5, and S1-S4 to reflect this update. The new calculations do mute the vapor flux and effectively force the latent heat to zero across all sites as expected using $T_{dew}$ as the snow surface temperature. We have updated our Results section to present our new relative contribution of the turbulent fluxes (section 3.3).

2. The formulae used to calculate incoming longwave radiation (equations 8 and 9) may have errors. Two errors related to this method were published in Table 1 of Flerchinger et al. (2009) for the Dilley and O'Brien (1998) method (their model B). Specifically, the parameter in the denominator at the end should be 2.5, not 25.

    - Dilley and O'Brien (1998) defined the parameter in the denominator at the end of their model B as $\omega^* = 25$ kg m$^{-2}$ much as we do in our formulation of the longwave energy balance equation, Eq. 8. (see the Dilley and O'Brien (1998) explanation of their model B on page 1394). Our formulation of the Dilley and O'Brien method is therefore correct.

Additionally, the precipitable water equation should have "465" instead of "4650" for w in centimeters, e_0 in kPa and T_0 in Kelvin. You should confirm with the original Dilley and O'Brien and Prata papers to ensure the errors from the Flerchinger table are not repeated here. Please check and revise calculations as needed. Careful description of all variables and units in the longwave equations will help others discern their accuracy, so please include.

    - Thank you for pointing out this error. The precipitable water equation in Prata (1996) has $w = 46.5 * (e\_0 / T\_0)$ with w in g cm$^{-3}$, e_0 in hPa, and T_0 in kelvin (Eq. 16, Pg. 1133). We used the value reported in Flerchinger et al., (2009) and have since corrected our equations accordingly. In rerunning our analysis with the correct value for *w* we found no significant difference ($< 0.1$ W m$^{-2}$) and this error did not affect our conclusions. However, we have modified all calculations within the results and discussion. We appreciate this noted error in the literature and we will not to perpetuate it going forward.

TECHNICAL CORRECTIONS

- P1, L27: You might also note here that melt rates may be decreasing with earlier snowmelt timing, based on the recent Musselman et al. (2017) paper.
    - Thank you. We have included this reference as it pertains to this research and it adds to the state of the science.

- P2, L09: I would avoid use of contraction "don't". Suggest replacing with "do not".
    - Corrected

- P3, L11: Add "on" after "focusing".
    - Corrected

- P3, L25: I would replace the phrase "principal components" with another one, so as to not confuse this with principal component analysis, a statistic analysis method. Perhaps "principal drivers of melt" is an appropriate replacement?
    - Changed to avoid confusion

- P4, L09: Add "to" after "sensitivity".
    - Added

- P4, L11: Remove "for" after "MRB".
    - Removed

- P4, L29: Perhaps state why the High sites were not included before WY2014? Were these installed after the Low and Mid sites?

    - That is precisely the reason why no High site data was included for WY2012 and WY2013. We have added a statement of clarity to address this comment.

- P5, L02-03: Please consider estimating the approximate uncertainty for the depth measurements, in terms of representing conditions with an interval of point observations. You could readily estimate using the graphs of Trujillo and Lehning (2015), assuming their errors estimates (made in continental zone) are valid in a maritime zone.

    – Trujillo and Lehning (2015) conclude that at 5m sampling intervals there is quite low uncertainty (Figure 6a). The normalized square error for an L of ~5 m was 0.3. We have added a statement in the text about the uncertainty of our sampling design.

- P05, L04: In your estimation of SDD, please state whether there were any late season snow storms after the final snow courses.

    – In our analysis we estimate SDD based on a linear interpolation using the median snow depths from the last two snow courses. Therefore, this measurement does not account for late season snow events. We added a statement of clarification to this point (Sec. 2.2).

- P05, L07: Please check whether the journal allows a single sub-section (i.e., 2.2.1 is presented, but not 2.2.2, etc.). Is this sub-section even necessary, or could it just be another paragraph in section 2.2?

    -We agree that this sub-sub section is unnecessary and have merged section 2.2.1 with section 2.2.

- P05, L13: Why was NLCD 2001 selected, when a more recent canopy cover product is available (i.e., NLCD 2011)? Please clarify to what degree did the forest change between 2001 and the study period.

    – The binary regression tree method was performed in 2010 and therefore the 2011 NLCD was not available at that time. We have cited Gleason et al., (2017) as they give a detailed overview of the methodology of selecting the Oregon ForEST site locations.

- P05, L19: Should Table 2 be cited here instead? Table 2 shows the tree characteristics. You might consider reordering the tables (swapping 1 and 2), such that they are introduced sequentially based on the narrative.

    – Thank you for pointing out the erroneously numbered tables. The error has been fixed.

- P05, L26: After "surface", consider adding a semicolon or starting a new sentence.

    – Added a comma after "surface" for clarity.

- P05, L30: Consider citing Friesen et al. (2014) here for a review of interception measurements and their limitations.

    – We have added this critical reference.

- P06, L02: Please clarify how you define an event. This may be tricky when there is intermittent snowfall through a day, for example.

- An event is defined at P05, L34 as "the daily increase of measured snow depth in the Open sites greater than 3 cm."

- P06, L17: Remove "is" before "cannot". Also, I would note that this can be measured using different sensors (e.g., IR temperature sensor), so you might want to change the phrasing of this sentence.
  – We agree that IR temperature sensors can directly measure the radiative temperature of snow, which can be converted to the kinetic temperature if the emissivity is known or assumed. We have reworded the phrasing to clarify this point.

- P07, L17: Remove "on" after "subsequently".
  - Removed

- P07, L18: Rephrase to say "Incorporating a sky-view factor (SVF) into the longwave radiation calculations allowed us…"
  - Rephrased for clarity.

- P09, L29: Please state which significance test was used.
  - We have rephrased the statement to provide clarity.

- P10, L10-11: As stated, this is a bit misleading. The topic of this paragraph is snowmelt rates, but it does not make sense that the melt rate is the reason why snow disappears earlier in Low-Forest relative to Low-Open. In fact, the melt rate is higher in the low-open than the low-forest. Hence, the longer lasting snow in low-open is likely related to snow accumulation dynamics (i.e., more interception losses in low-forest), something that is recognized in the discussion section (P13, L24-26). Please consider restating to avoid confusion.
  – We agree and have clarified this statement so as to avoid any confusion.

- P10, L12: Presumably, high-forest is being compared to high-open here, but that is not explicitly stated. Please consider including this.
  –This is added for clarity.

- P10, L16: "duration" of what? Snow in the canopy?
  –We have added the study duration years used in this calculation.

- P10, L22-30: These reported R2 values do not match what is shown in Figure 3. Should they?
  –The reported values represent the $R^2$ for all qualifying events, e.g. those both less than 15 cm and greater than 15 cm. Within Figure 3 we identified a threshold of 15 cm where a trend emerges and reported those $R^2$ values within the figure. We have clarified this within the text.

- P11, L10-16: This is more appropriate for the discussion section, rather than the results section. Also, some useful context would be the frequency and severity of rain on snow events at the study sites during the study period.
  – We feel these few lines provide context for our longwave calculation results since these sites are influenced by ROS events. However, we did not measure total precipitation and would not be confident in estimating ROS frequency/intensity without a larger effort that is beyond the

scope of this paper.

- P13, L25: "it is" instead of "it's".
    - Amended.

- P14, L30: Replace "effects" (noun) with "affects"( verb).
    - Replaced.

- P14, L30: Remove "truly" (avoid adverbs in science writing).
    - Removed.

- P15, L03: Remove "the" before "emerged".
    - Removed.

- Fig. 1: The inset map is of low quality. Text and markers are too small to read, and this may not be comprehensible to someone unfamiliar with the region. Can this be improved? – We have swapped out the Columbia River Basin inset for a basemap of the USA. This provides more spatial context for a wider audience.

- Table 2: Consider including variability in the tree characteristics (e.g., report the standard deviation as a plus/minus next to each average value). This could be useful context. We agree and have added the variability to Table 1.

- Tables 1 and 2: I do not understand why C_IE at high-forest is 31% for the full study duration, when only one year of C_IE at high-forest is shown in Table 1 (39%). Either this is an error (and both should be 39%), or the study duration also includes another year (WY2015, likely C_IE~23%)? If the latter, then why is WY2015 not included in Table 1? Please check and clarify. – This was a simple error on the part of the primary author and has been corrected.

- Figures 4, 7: Can you please clarify whether the range in the energy balance is across the water years?
    – In Figures 4 and 7 the figure caption notes WY 2012 – WY 2015 for the range.

- Figure 5: The colors for longwave and sensible heat are difficult to distinguish. Also, the narrow widths of the bars make this figure difficult to inspect. Consider highlighting specific, shorter periods rather than the entire water year.
    – Depicting an entire year is helpful to illustrate the variability of energy fluxes (in the forest) and how the primary energy contributor changes within the forest as the spring progresses. Within the text we discuss in depth a certain period within the water year that highlights a large change in the data, something that would not be easily distinguished if we "zoomed in" to a specific period within the water year.

BIBLIOGRAPHY

[revised manuscript text omitted]

---

## Author Response (AR3)

**Editor comments:**
Thanks for your revisions, I find the manuscript now almost ready for publication. Below I list a few minor points which should be rather straight forward to address.

1) you use a lot of abbreviations, perhaps a list of abbreviations could be helpful? At least, please make sure to explain all abbreviations, as far as I can see, the description for SDD is missing

2) please follow the author guidelines with respect to the equations and avoid multi-letter variable names such as SR in eq 3, similar in eq 6&7, and the use of a x as multiplication sign

3) p10, you discuss the threshold and linear correlation, Perhaps computing rank correlations could be a good way to further evaluate this

4)p11, l13-19, these lines are no results, move to intro or discussion

5) p12l35, some word (verb) seems to be missing here

**Authors response:**
1) We have included a supplemental table that includes all abbreviations in the manuscript with a brief definition of each (Table S4). SDD was defined on pg. 5 line. 7.
2) We have updated the manuscript to avoid multi-letter variable names and changed all multiplication signs to the correct notation.
3) We have included a Spearman's rank correlation analysis statement into the manuscript (pg. 10 lines 28-30).
4) We have moved these lines into the Discussion (pg. 12 lines 9-15).
5) We have modified this sentence for clarity.